# The fly connectome reveals a path to the effectome

Dean A. Pospisil[1,5 ✉], Max J. Aragon[1,5 ✉], Sven Dorkenwald[1,2], Arie Matsliah[1], Amy R. Sterling[1], Philipp Schlegel[3,4], Szi-chieh Yu[1], Claire E. McKellar[1], Marta Costa[4], Katharina Eichler[4], Gregory S. X. E. Jefferis[3,4], Mala Murthy[1] & Jonathan W. Pillow[1]

A goal of neuroscience is to obtain a causal model of the nervous system. The recently reported whole-brain fly connectome[1–3] specifies the synaptic paths by which neurons can affect each other, but not how strongly they do affect each other in vivo. To overcome this limitation, we introduce a combined experimental and statistical strategy for efficiently learning a causal model of the fly brain, which we refer to as the 'effectome'. Specifically, we propose an estimator for a linear dynamical model of the fly brain that uses stochastic optogenetic perturbation data to estimate causal effects and the connectome as a prior to greatly improve estimation efficiency. We validate our estimator in connectome-based linear simulations and show that it recovers a linear approximation to the nonlinear dynamics of more biophysically realistic simulations. We then analyse the connectome to propose circuits that dominate the dynamics of the fly nervous system. We discover that the dominant circuits involve only relatively small populations of neurons—thus, neuron-level imaging, stimulation and identification are feasible. This approach also re-discovers known circuits and generates testable hypotheses about their dynamics. Overall, we provide evidence that fly whole-brain dynamics are generated by a large collection of small circuits that operate largely independently of each other. This implies that a causal model of a brain can be feasibly obtained in the fly.

A fundamental barrier to resolving a causal model of the nervous system is that causal relationships in the brain cannot be inferred solely from passive measurements of neural activity[4,5]. Direct perturbation of neural activity (for example, optogenetic stimulation) confronts this problem and has therefore been an area of intense methodological research and resulting progress. However, a clear approach for how to use these tools to obtain a causal model of neural activity has not emerged. Here we introduce a combined statistical and experimental strategy and demonstrate that it can efficiently learn a causal model of the fly brain.

We adapt a technique known in the statistical literature as 'instrumental variables'[6] (IVs). This technique was developed to estimate causal relationships in observational data in which direct experimental control is unfeasible. It relies critically on the stringent requirements of an IV: that it only directly affects an observed variable that putatively affects the outcome of interest—in the absence of that effect, the IV is independent of all variables. The observed relationship between the IV and the outcome is then strictly a result of the causal effect of interest. Yet, despite the stringency of these requirements, optogenetic stimulation plausibly meets them[7,8]. Optogenetic stimulation affects neural activity, is independent of neural activity because it is controlled by the experimenter, and acts on the brain only through neurons that express opsins. Thus, the IV approach could in principle be used to estimate causal effects between neurons.

However, there are two fundamental problems with the IV approach applied to the entire nervous system of the fly. First, naively estimating effects between every pair of neurons in the fly (approximately $(10^5)^2$ pairs) would require intractable amounts of data. This problem is an even more insurmountable barrier to learning causal models of organisms with larger numbers of neurons (for example, $(10^8)^2$ potential effects for mice[9]). Second, it would be unfeasible in the fly—and in most organisms—to independently stimulate and record from all neurons at once. The effectome would thus need to be gradually constrained across experiments on small sub-populations. It is unclear how to order experiments such that insights into whole-brain dynamics are achieved efficiently. Here we show that the Flywire connectome[1–3] provides a feasible path to surmounting both barriers in the fly.

First, a principled approach to improving the data efficiency of an estimator is to place priors on the parameters being estimated. The fly connectome can be used as a prior on effects between neurons in the fly brain: neurons with no synaptic contacts are unlikely to directly affect each other. The connectome of the fly is exceedingly sparse (around 0.01% of neuron pairs form a synaptic contact[10]): thus, a strong prior can be placed on the vast majority of interactions in the effectome. Furthermore, with synaptic counts and synapse-level electron microscopy-based neurotransmitter predictions provided by the Flywire connectome[11] (see Methods, 'Construction of fly connectome

[1]Princeton Neuroscience Institute, Princeton University, Princeton, NJ, USA. [2]Computer Science Department, Princeton University, Princeton, NJ, USA. [3]Neurobiology Division, MRC Laboratory of Molecular Biology, Cambridge, UK. [4]Drosophila Connectomics Group, Department of Zoology, University of Cambridge, Cambridge, UK. [5]These authors contributed equally: Dean A. Pospisil, Max J. Aragon. ✉e-mail: dp4846@princeton.edu; mjaragon@princeton.edu

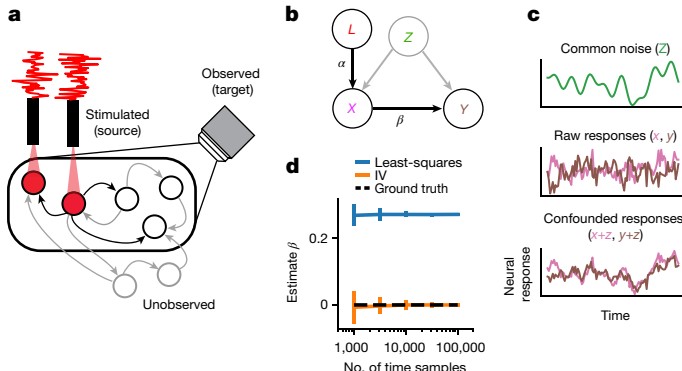

**Fig. 1 | Schematic and simulation of effectome inference using IVs. a**, Example of a single fly experiment. Some sets of neurons in the fly are observed (black circles within the field of view of the microscope), and a subset of these express opsin and are being stimulated (neurons under red light) with a white noise pattern (red trace). Another set of neurons may be unobserved. All neurons can have directed interactions via synapses (arrows), and only the effects of stimulated neurons can be estimated (black arrows connecting stimulated neurons to observed neurons). **b**, In this setup, the laser can be cast as an instrumental variable (IV) that directly affects only the opsin-expressing neurons (arrow labelled $\alpha$ from $L$ to $X$). Stimulated neurons in turn have a direct effect on downstream observed neurons (arrow $\beta$ from $X$ to $Y$), but common unobserved inputs ($Z$) may corrupt attempts to estimate the direct relationship. The IV approach uses the joint relationship between the laser and downstream neurons to determine the direct effect of the stimulated neurons on postsynaptic neurons. **c**, Top, a simulated example of confounding effect $Z$, which is given by a slow drifting signal (smooth green trace). Raw, uncorrupted responses of neurons $X$ and $Y$ are independent (pink and brown traces, middle), meaning that there is no connection from $X$ to $Y$. Bottom, observations $X$ and $Y$ after adding the confounding signal $Z$, resulting in substantial correlation. **d**, The least-squares estimate of the weight from $X$ to $Y$ (blue line; mean ± s.d.; $n = 100$ simulations) exhibits large bias regardless of sample size, whereas the IV estimate (orange line) converges to the true effectome weight of zero (black dashed line).

matrix'), plausible priors can be placed on the sign and magnitude of effects between the neurons that are connected (the confidence of those predictions can naturally be incorporated into the strength of the prior).

Second, because the connectome is a complete connectivity map of the fly brain, it is uniquely suited to guide efficient estimation of the effectome. Neuroscientists tend to study a given neural circuit because it has been studied previously. A data-driven approach to proposing neural circuits of interest could reveal important computations that had not yet been considered. Ideally such a method would discover independent neural circuits and rank them by their total effect on the brain. To predict which neural interactions contribute most strongly to whole-brain dynamics, we analyse the eigenmode decomposition of the connectome. The eigenvectors of the connectome, ordered by the magnitude of their eigenvalues, provide sets of neurons, and patterns of neural activity therein, that under our 'connectome prior' are predicted to have the greatest total effect on the brain.

We thus provide an experimentally tractable paradigm for learning a complete causal model of the fly brain that is uniquely enabled by the properties of the whole-brain connectome. Specifically, we show that the sparse connectivity between neurons markedly improves the efficiency of estimating causal effects. We also show that that small populations of neurons underlie dominant dynamical modes, suggesting that whole-brain dynamics can be constrained with a series of highly targeted—and thus feasible—experiments.

Here we first lay out an experimental setup and associated statistical model of neural activity in the fly. Next, we outline how to use optogenetic perturbation as an IV to infer the direct causal effects between

neurons in the context of a linear dynamical system. We then show how, in the presence of confounds, the classic regression estimator will return biased results, whereas the IV approach gives a consistent estimator. Using a simulation of whole-brain neural activity based on the connectome, we demonstrate that the IV approach provides consistent estimates of ground truth neural connectivity. We find that because there is a massive number of potential downstream neurons from any given neuron, the standard IV regression estimator converges very slowly. This motivates use of the connectome to formulate a prior on the IV weights so that the estimator remains consistent (that is, even if the prior is wrong, the estimator will converge to ground truth with enough data) but orders of magnitude more efficient—to the degree that the prior is correct. Finally, we analyse our 'connectome prior' to reveal thousands of proposed circuits ranked by their predicted total effect on the brain. We analyse two of these circuits and find that one recapitulates a proposed circuit for computing opponent motion and the other provides a dynamical mechanism for visual spatial selectivity.

Our motivating setting is an optogenetic experiment in the fly (Fig. 1a): the activity of a population of 'target' neurons is observed, a subset of these are 'source' neurons that express opsin driven by $n_l$ independent lasers, and the remaining neurons remain unobserved—these can also be unobserved non-neuronal processes recurrently interacting with the observed neurons.

The graphical model that we associate with this setup identifies the lasers as the IVs that independently drive the source population ($X$) via a linear transformation ($\alpha$), the source population drives the target population ($Y$) via $\beta$ (the effect), and both receive common inputs from unobserved confounders $Z$ (Fig. 1b). The fundamental difficulty of fitting neural models to observed data is that even if $\beta = 0$, $Z$ can induce spurious dependence between the source and target population.

## IVs are robust to unobserved variables

The principal challenge to inferring causal effects solely through passive observation of neural activity is the possibility of unobserved confounding inputs. There are many potential unobserved confounders in the fly. Typically, a small subset of the nervous system is imaged at a time: thus, input from unobserved neurons could confound the observed neurons. Whole-brain imaging is possible in the fly[12,13], but resolving single-neuron dynamics remains a challenge owing to the density of the neuropil. Even if single neuron activity could be resolved, afferent activity from the peripheral nervous system and ventral nerve cord could potentially be a source of common variability. Moreover, neuropeptide signalling cannot be inferred from calcium imaging data alone[14,15]. Finally, measurement error, including physiological artifacts such as brain movement[16], can act as a source of common variability. Collectively, there is a high probability of unknown confounding variables preventing valid causal inference during passive observation of neural activity in the fly brain.

A simple case that demonstrates the effect of confounding variables is one in which two neurons $X$ and $Y$ have no causal effect on each other ($\beta = 0$ between $X$ and $Y$; Fig. 1b), but there is a common unobserved input to both from $Z$ (for example, an unobserved neuron). Even though there is no causal effect of $X$ on $Y$ (Fig. 1d, ground truth weight is 0), the least-squares estimate incorrectly converges on a positive weight, whereas the IV converges on 0 because there is no correlation between the laser and $Y$. Thus the IV, under our assumed model, is not corrupted by unknown, unobserved inputs.

## IV accurately estimates fly effectome

To demonstrate that the IV can, in principle, estimate the fly effectome, we applied it to a simulation of the entire fly brain during stimulation of a single source neuron and a whole-brain recording (see Methods, 'Simulations to evaluate estimators'; for a graphical model, see

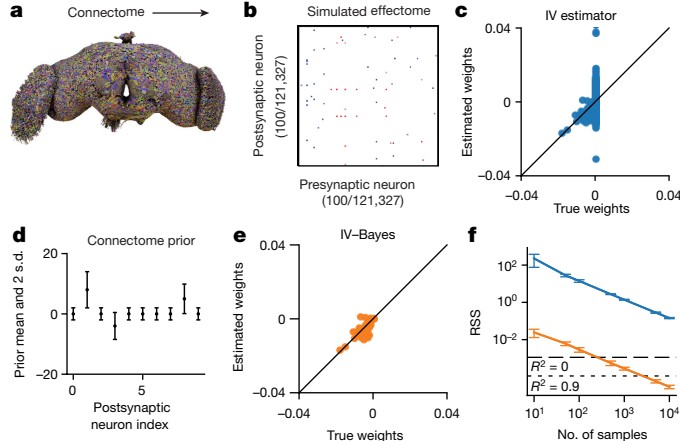

**Fig. 2 | Inferring the effectome using standard and Bayesian approaches on simulated data. a**, We used the connectome to set the effectome weights for a whole-brain simulation using all 121,327 connected neurons (neurons with no incoming or outgoing connections, above a threshold of >5 synapses, were not included). **b**, Synaptic weights were set proportional to synapse count, with positive (negative) sign for excitatory (inhibitory) synapses. **c**, IV estimates of postsynaptic weights of a single example neuron. Most of the error falls along the vertical line where true weight equals 0 (which is the majority of the weights, owing to the sparsity of the fly connectome). **d**, Mean ± 2 s.d. of an independent Gaussian connectome prior on each weight in the effectome. We set the prior mean to be proportional to the signed synapse count of the connectome and variance equal to the absolute value of the mean plus a small constant, so that the prior width is non-zero between neurons with no known synapses. **e**, An IV–Bayes estimator shows smaller error than the raw IV estimator in **b**. **f**, The error of estimated weights (residual sum of squares, RSS) decreases with the number of samples for both estimators, but IV–Bayes gives several orders of magnitude faster convergence (orange below blue line; mean ± s.d.; $n = 10$ simulations). Mean squared error will decrease indefinitely for both estimators because they are consistent (that is, they converge to ground truth as the number of samples goes to infinity). Horizontal lines show error level where the $R^2$ of the recovered weights is zero (long dash) and 0.9 (short dash), respectively.

Extended Data Fig. 1). We set our ground truth causal effect matrix entries to be proportional to the number of synapses multiplied by the 'sign' of those synapses—whether they were inhibitory or excitatory (Fig. 2a). We found that the estimator on average was accurate (Fig. 2b, scatter centred around diagonal), but the bulk of error came from estimates of weights whose true value was zero. This was because the vast majority of weights are zero owing to the sparsity of the fly connectome. Non-zero estimates of these weights were thus the dominant contribution to total estimation error.

## The connectome as an effectome prior

The high variability of IV reflects a fundamental problem in fitting a model of the entire brain: the number of parameters is large. Yet, because the fly connectome is available and—critically—it happens to be highly sparse, we can markedly increase the efficiency of our model estimate by assuming that neurons with no synaptic contacts are unlikely to directly affect each other. We do so in a principled manner with an extension of our estimator to a Bayesian setting.

We took a Bayesian approach to reduce error by placing a Gaussian prior on the model weights (IV–Bayes). The Gaussian prior mean was proportional to the synaptic count and sign (positive for excitatory synapses and negative for inhibitory synapses) (Fig. 2d). The variance was equal to the absolute value of the mean plus a small constant. Thus, weights between non-anatomically connected neurons are strongly biased towards zero, whereas weights for connected neurons are weakly biased toward the scaled synapse count, allowing them to be zero if

warranted by the data. We add a small constant to the prior variance so that the estimator remains consistent: even if the prior is wrong (for example, if a synapse exists where none was found in the connectome), the estimator will converge to ground truth with enough data.

We evaluated the IV–Bayes estimator using a simulated dataset generated with effectome weights set to a corrupted version of the ground truth connectome weights, thus creating a mismatch between the effectome weights and the prior mean (Methods, 'Simulations to evaluate estimators'). This corruption could, for example, reflect natural variation between flies' connectomes. We note that the IV estimator is not meant to estimate the underlying connectome. Instead, the estimator approximates the linear effects between neurons—these effects may have a weak relationship with the connectome, and will probably depend on the state of the nervous system (see Discussion, 'A broader definition of the effectome'). We found that the IV–Bayes estimates outperformed the standard IV estimator, even though the prior mean was corrupted (Fig. 2c,e). In particular, the high variability of the IV estimate for zero weights was quenched in the IV–Bayes estimate. Intuitively, if the connectome provides information about the strength of causal interactions between neurons, it should outperform standard IV.

To quantify the relative efficiency of the naive IV approach and IV–Bayes, we sampled effectome matrices as described above and then evaluated the average residual sum of squares (RSS) of the two estimates as a function of the number of samples (for example, duration of experiment). As expected, we found that the RSS of both estimators decreased with increasing samples (Fig. 2f, blue and orange trace slope downwards). Yet, we found that the RSS of the IV estimator is at least an order of magnitude higher than that of the IV–Bayes estimator across number of samples (blue above orange). In terms of fraction of variance explained, IV–Bayes explains the vast majority of variance for the maximal number of time samples (orange trace below dotted line on right) but the raw IV estimator is still too noisy to achieve a positive quantity of fraction variance explained (blue trace above dashed line) (Fig. 2f). Thus, in simulation, IV–Bayes provides at least an order of magnitude improvement in converging to the ground truth causal effects.

Our simulations thus far have focused on the stimulation of a single neuron while the entire fly brain is observed. If every neuron in the fly brain was stimulated independently while every neuron was being observed, the entire fly effectome would be identifiable within a single experiment (Extended Data Fig. 2b; for multi-neuron simulations see Extended Data Figs. 3–6). However, it is unclear whether this approach is experimentally feasible, given the constraints of diffraction-limited optics, which makes independent stimulation and identification of every neuron in a whole-brain imaging setup challenging. A more feasible approach would be to sparsely image and stimulate neurons to estimate the effectome across flies (Extended Data Fig. 2c). We now describe a data-driven strategy for systematically choosing subsets of neurons that account for disproportionate shares of neural dynamics.

## The connectome reveals dominant circuits

Here we demonstrate a data-driven method for ranking sets of source neurons that are most likely to form circuits with a large effect on the fly nervous system. We propose that these circuits should be prioritized for interrogation by our estimator. Specifically, we consider a recurrent neural network model of whole-brain activity given by

$$\mathbf{r}_{t+1} = W\mathbf{r}_t, \tag{1}$$

where $\mathbf{r}_t$ is a vector denoting the activity of all neurons at time $t$ and $W$ is the effectome weight matrix, which (for these analyses) we set to the scaled, signed synaptic counts extracted from the connectome. To analyse the fly brain's dynamical properties, we perform an eigen-decomposition of the weight matrix $W$, which decomposes global dynamics into patterns of neural activation called eigenvectors with

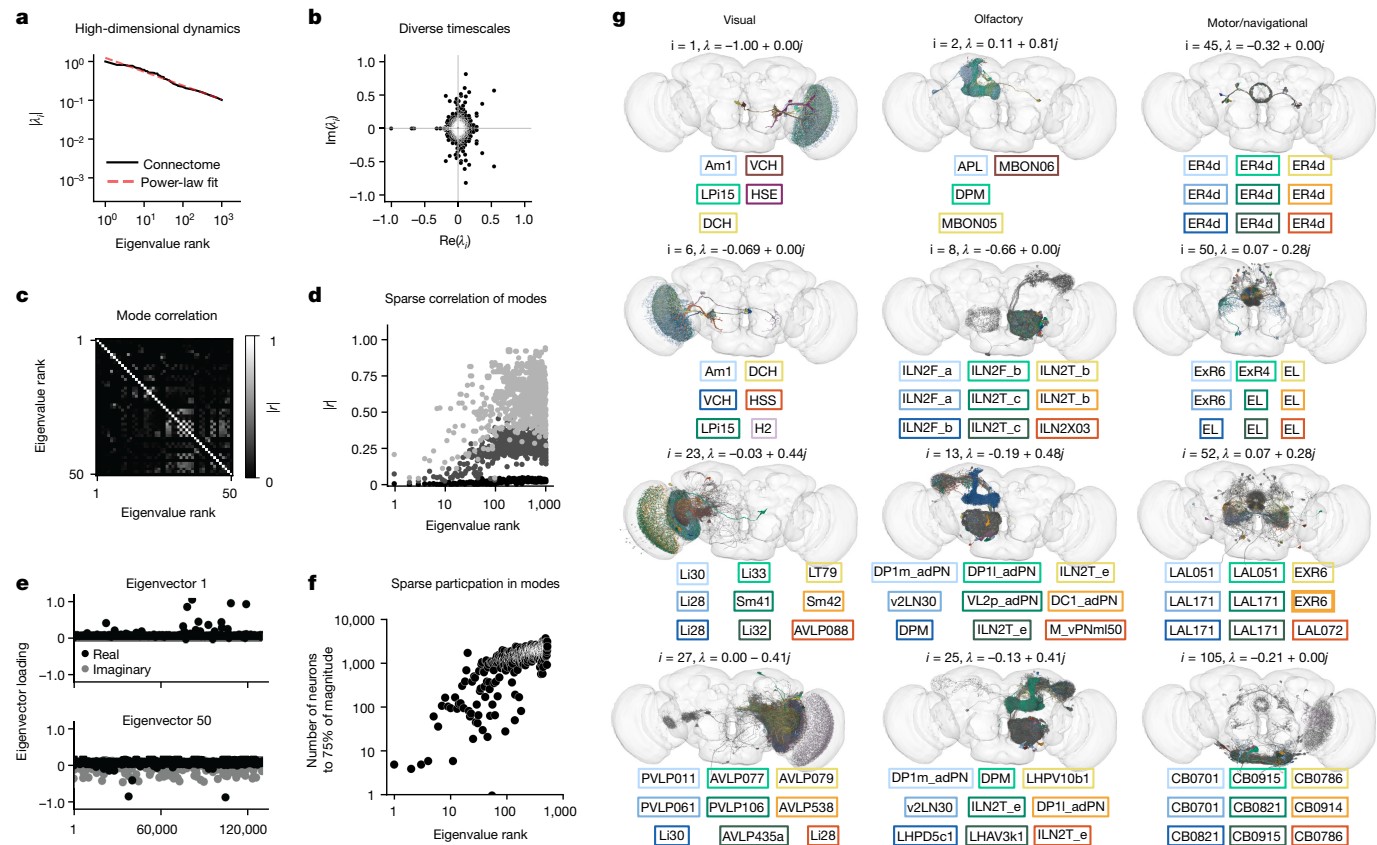

**Fig. 3 | Putative global dynamical properties of the fly central nervous system. a**, Magnitude of the top 1,000 eigenvalues of the putative effectome (scaled matrix of signed synaptic counts extracted from the connectome) and power-law fit. **b**, Eigenvalues plotted in the complex plane. **c**, Correlation of eigenvectors sorted by associated eigenvalue magnitude. **d**, Median (black), 99th percentile (dark grey) and maximum (light grey) correlation between each eigenvector and all other top 1,000 eigenvectors, showing that the first 10–20 eigenvectors are nearly orthogonal to other eigenvectors and the correlation between other eigenvectors is highly sparse. **e**, Per neuron eigenvector loadings for the first (top) and 50th (bottom) eigenvector. **f**, Concentration of eigenvector loadings, quantified by the number of neurons needed to account for 75% of the power of the eigenvector. **g**, Anatomical renderings of neurons needed to account for 75% of the eigenvector's loading power. Eigenvectors are drawn from the top 100 eigenvectors, and each column displays eigenvectors with neurons predominantly associated with either visual, olfactory or motor/navigation anatomical locations.

simple two-dimensional rotational dynamics determined by an associated eigenvalue. Below, on the basis of this eigendecomposition, we provide testable hypotheses pertaining to both global properties of fly neuronal dynamics and highly specific circuits.

In the model we have proposed (equation (1)), the eigenvectors of the effectome describe the dominant modes or patterns of neural activity that will grow or decay over time, each governed by its associated eigenvalue. For example, if the activity of the brain at time 0 is set to the $i$th eigenvector $\mathbf{v}_i$ (whose eigenvalue is $\lambda_i$), then the brain's activity pattern at time step $t$ is given by

$$\mathbf{r}_t = \lambda_i^t \mathbf{v}_i. \tag{2}$$

Thus, the magnitude of the eigenvalue precisely determines the magnitude and duration of the effect of this pattern of activity. The eigenvectors with the largest eigenvalues are therefore plausibly associated with neural dynamics that have the largest total effects on the fly brain. The neurons associated with the significant coefficients, or 'loadings', in an eigenvector indicate the sub-population of neurons whose connectivity principally sustains these dynamics, forming an 'eigencircuit'.

There are two critical properties of the eigendecomposition that determine the rate at which neural dynamics can be constrained by the estimated effectome. The first is the sparsity of eigenvectors. If the pattern of activity specified by the eigenvector includes only a few neurons with non-zero loadings, then only the effectome of those neurons needs to be learned to specify that dynamical mode. By contrast, if each eigenvector significantly involves all neurons, the entire effectome would need to be learned to explain even one dynamical mode. The second critical property is the relative magnitude of the eigenvalues. If the eigenvalue associated with a sparse eigenvector was much larger than all the others, then the majority of variation in global dynamics could be explained with the effectome of a handful of neurons.

## Global dynamics of putative effectome

We first examined the relative magnitude of the eigenvalues and found they decayed slowly. For example, the 1,000th eigenvalue has approximately 1/10 of the magnitude of the largest (Fig. 3a). This implies that: (1) the choice of which early modes to analyse is somewhat arbitrary, because they have similar magnitudes; and (2) many dynamical modes could be required to explain neural dynamics in the fly brain (that is, fly neural dynamics are high-dimensional). We note that the dimensionality of neural dynamics depends on the input distribution. This analysis implicitly assumes private white noise inputs to each neuron where all eigenvectors are equally likely to be driven—intuitively, this may correspond to a resting state with each neuron stochastically firing at a similar rate.

We explored the range of timescales for predicted dynamics by examining the eigenvalues in the complex plane (Fig. 3b). Complex

eigenvalues are associated with a complex eigenvector whose real and imaginary parts define a plane of rotational dynamics for neural activity. If a pattern of activity is induced within this plane, then neural activity will continue to evolve solely within the plane—that is, it will transition over time between mixtures of the real and imaginary part of the eigenvector. The angle from the positive real axis determines the speed of these rotational dynamics. At 0°, the eigenvalue is real positive and there are no oscillatory dynamics but a simple monotonic decay. A non-zero angle is exactly the angular step size of rotational dynamics at each time step, so small angles imply slow rotational dynamics, and a negative real eigenvalue (180°) implies the fastest possible rotational dynamics (flipping sign at each time step). We find there is a broad distribution of timescales of rotational dynamics. The distribution of angles had modes at 0°, 90° and 180°, implying a preponderance of monotonically decaying modes, rapid transitions between distinct populations and the fastest possible timescale, respectively.

We then investigated whether dynamical modes tended to be independent of each other. When eigenvectors are orthogonal to each other, the dynamics associated with each eigenvector are independent (under assumed white noise inputs). Conversely, if two eigenvectors are highly correlated, then these dynamics are more likely to co-occur. Thus, by examining the correlation between eigenvectors we can test which dynamical motifs will tend to be enlisted simultaneously—potentially because they are involved in similar computations. Examining the correlation matrix of the first 100 eigenvectors (treating the real and complex parts as separate eigenvectors), we found on average correlation was weak with sparsely distributed higher values (Fig. 3c). We observed that early eigenvectors tended to have lower correlation to others (first ten rows and columns dark). Aggregating the correlation of each of the top 1,000 eigenvectors to all others, we found a clear trend in which both the max and average correlation increased with eigenvalue rank (Fig. 3d). Roughly speaking, the top ten dynamical modes will occur independently of each other, whereas the rest will tend to co-occur with at least one other mode. Thus, we predict that dynamical modes will typically operate independently of each other, which bodes well for the project of examining these circuits individually.

Experimental limitations dictate that only sparse populations of neurons can be simultaneously identified with the connectome and independently stimulated. Thus, here we analysed whether the populations involved in these eigenmodes are in fact sparse. Plotting the first eigenvector, we found that the loadings across neurons were indeed extremely sparse, with most loadings near zero and only several deviating significantly from zero (Fig. 3e, top). These loadings were of the same sign, indicating that all neurons significantly involved in this mode oscillate in sync. We found that later modes were also sparse, but less so (Fig. 3e, bottom). We measured the number of neurons required to account for 75% of power across loadings for the first 1,000 modes, and found that this number never exceeded more than 10% of the fly brain (Fig. 3f). The dominant modes tended to be the sparsest, suggesting that the dominant dynamics of the fly brain can be explained by estimating only a small fraction of the effectome. On average, for the top 10 eigenvectors, around 50 neurons were needed (less than 0.05% of all neurons), for the 10th to 100th eigenvectors, around 500 neurons were needed (less than 0.5% of all neurons), and for the 100th to 1,000th eigenvector, around 1,500 neurons were needed (less than 1.25% of all neurons). These findings suggest that the dynamics of the dominant modes in the fly brain can be explained by estimating a small fraction of the effectome.

Finally, we characterized the anatomical properties of these putative circuits. We visualized up to 100 neurons that together comprise 75% of loading power within their respective eigenvector loadings and colour-coded the top 12 (the remaining neurons are in grey; by contrast, all results in Fig. 3a–e are for all neurons). To provide a broad sampling of circuits, we organized eigenvectors into three groups on the basis of

anatomical location: visual, olfactory and motor/navigational (Fig. 3g, left, middle and right columns, respectively). In general, we found that the highly sparse top eigenvectors that we previously characterized were also anatomically localized. For example, the top visual eigenvectors contained neurons that were confined to the lobula plate in the left hemisphere (row 1) and right hemisphere (row 2) (Fig. 3g). These eigenvectors recapitulate a hypothesized neural circuit for opponent motion computation (Fig. 4a–f). The top olfactory eigenvectors were also anatomically localized and contained mushroom body neurons (row 1) and projection neurons from the antennal lobe to the lateral horn (row 2). Multiple motor/navigational eigenvectors also showed confinement to the ellipsoid body (rows 1 and 2). For all three anatomical categories, we observed that eigenvectors with lower sparsity tended to incorporate diverse cell types distributed across multiple neuropils (rows 3 and 4). In general, we found that neurons with high loadings in early eigenvectors were often anatomically localized in accordance with the classical approach of studying the nervous system region by region. Conversely, many circuits were not anatomically localized and merit further investigation (for examples, see Extended Data Fig. 7).

We observed that anatomically localized eigencircuits were in the minority (10% of the top 1,000 eigencircuits), and non-localized circuits were often among the most dominant (Extended Data Fig. 8e,f). We considered whether modest amounts of biological variability, measurement error or saturation would change these results (Extended Data Fig. 9), and found that in general the early eigencircuits were quite robust, and the later ones less so. The non-local circuits tended to be less robust to perturbation, but we did find many robust examples (Extended Data Fig. 9b). We also determined that the high dimensionality of the connectome was not simply a result of the sparsity of the connectome, nor was it sensitive to the aforementioned factors (Extended Data Fig. 10).

We found that the top eigenvectors of the connectome prior are highly sparse. This property facilitates learning the effectome because it is easier to identify, image and stimulate sparse populations. Given the thousands of existing genetic driver lines accessible to the fly community[17] and tools for automatically screening these lines for neurons of interest[18], it may be possible to identify reasonably sparse genetic lines that contain a subset of each eigencircuit's dominant neurons (see Supplementary Information, 'Sparse expression'). Altogether, owing to the distinct properties of the fly connectome—namely the sparsity of connections, eigenvector loadings and interaction between eigenmodes—there exists a plausible path forward to systematically and efficiently explain whole fly brain dynamics in terms of direct causal interactions between neurons.

## Dominant circuits are interpretable

Our decomposition of the putative effectome revealed sparse eigenvector loadings, which makes them amenable to further analysis. We tested whether these vectors correspond to identifiable circuits and interpretable dynamics in the fly brain (see Methods, 'Simulations to analyse eigencircuits').

We found that the first eigenvector localized to the lobula plate (Fig. 4a) and was highly sparse (Fig. 4b). We quantified the anatomical localization as the fraction of synapses in a single neuropil. For this eigencircuit, 75% of synapses were localized to the lobula plate, whereas the rest were confined to the inferior posterior slope. The associated real eigenvalue was negative: thus, for linear dynamics we expect rapid oscillation, but in the more realistic case where activity is thresholded (see Methods, 'Simulations to analyse eigencircuits'), these neurons will inactivate rapidly following activation (Fig. 4c). The top four neurons in this eigenvector were VCH, DCH, LPi15 and Am1[19]. All putative effects between these neurons are inhibitory, but there is a complex mix of mutual and directed inhibition (Fig. 4d, partial symmetry across diagonal of weight matrix). Notably, VCH and DCH do not inhibit each

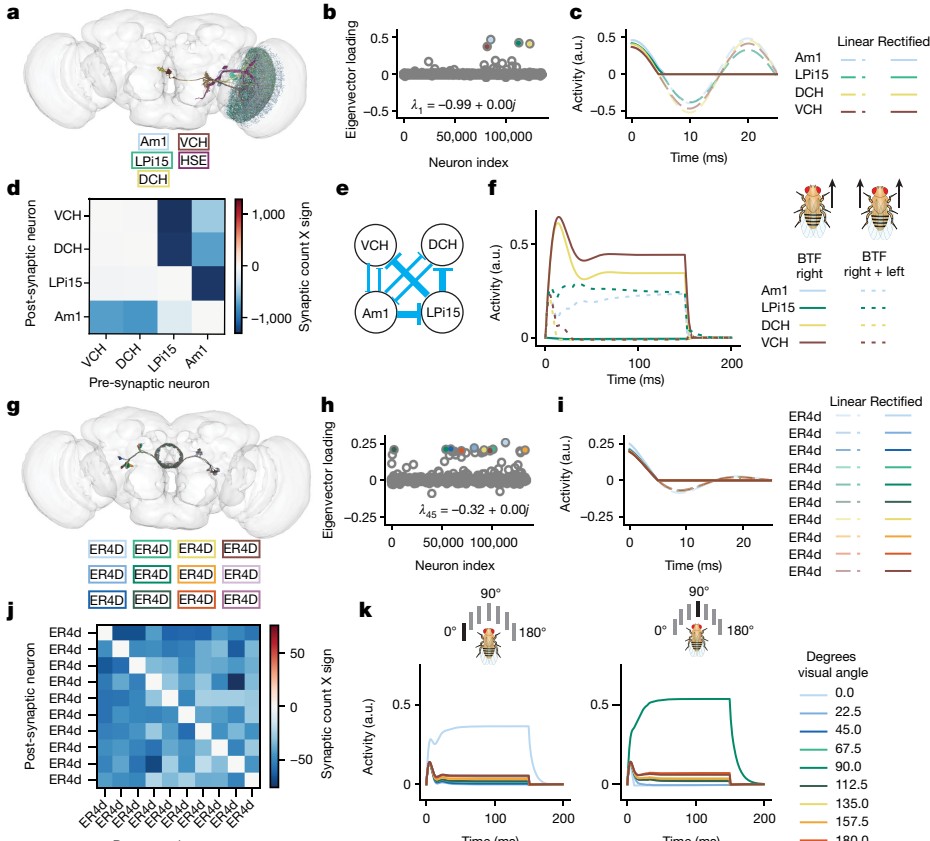

**Fig. 4 | Example eigencircuits. a**, Neurons with top five loadings on eigenvector 1. **b**, First eigenvector of the fly effectome. **c**, Linear and rectified dynamics upon stimulation by first eigenvector (a.u., arbitrary units). **d**, Synaptic count and sign between neurons. **e**, Circuit diagram representation of synaptic weight and sign. **f**, Visual stimulation simulation. VCH and DCH are given sustained stimulation for 150 ms, simulating BTF motion on the right side of the fly (fly on the left; solid brown and yellow, high sustained response). All neurons were given sustained stimulation for 150 ms, simulating BTF motion on both sides of the fly, which leads to inhibited responses of VCH and DCH (fly on the right; dashed brown and yellow below solid). **g**, Neurons with top

four loadings on eigenvector 45. **h**, All loadings for eigenvector 45. **i**, Linear and rectified dynamics (left) upon stimulation by eigenvector 45 (right). **j**, Synaptic count and sign between nine neurons with top loadings on eigenvector 45. **k**, Simulation of visual stimulation. Left, simulation of a 0° visual stimulus, with strong input to the 0°-preferring neuron (light blue trace) and sustained background input to all other neurons. The network response exhibits WTA dynamics, in which all neurons respond transiently to stimulus onset, but only the neuron with maximal response remains active for the entire stimulus duration. Right, similar results for a 90° stimulus and 90°-preferring neuron (green trace).

other, but are inhibited by Am1 and LPi15, but Am1 receives recurrent inhibition from VCH and DCH, whereas LPi15 does not (Fig. 4e).

Concretely determining the computation that this visual circuit may perform requires an assumption about how visual features drive these neurons. It is known that VCH and DCH receive major input from H2 from the contralateral eye, which responds to back-to-front (BTF) motion[20], whereas LPi15 and Am1 receive major inputs from T4b and T5b, which are driven by ipsilateral BTF motion[21]. We probed the functional properties of this circuit by simulating BTF visual input (see Methods) to either the right eye (contralateral input) or the left and right eyes together (bilateral input). When provided with contralateral BTF input only, we observe that VCH and DCH activity remained high throughout the stimulation period, whereas LPi15 and Am1 activity were suppressed because they were not directly stimulated and because Am1 is strongly inhibited by VCH and DCH (Fig. 4f, solid trace brown and yellow above blue and green). Conversely, bilateral BTF stimulation resulted in suppression of VCH and DCH. Overall, this putative circuit is well-suited to compute opponent motion across the fly eyes.

Notably, this circuit was analysed in a very recent small-scale connectomic analysis of the optic lobe[21], whereas it was 'rediscovered' with our data-driven approach. This suggests, anecdotally, that the eigendecomposition of the connectome can reveal scientifically interesting sub-circuits.

Inspired by our findings with the first eigenvector of the connectome prior, we next sought to identify additional circuits with a putative role in stimulus selection. We found that eigenvector 45 contained high loadings for GABAergic (γ-aminobutyric acid-producing) R4d ring neurons in the ellipsoid body (Fig. 4g). For this eigencircuit, 99% of synapses were localized to the ellipsoid body, and the remainder were in the fan-shaped body and mushroom body medial lobe. R4d ring neurons have spatial receptive fields that retinotopically tile the visual field and exhibit directional and orientation tuning[22]. This eigenvector has sparse and bimodal loading (Fig. 4h, majority of scatter near 0 but subset near 0.2). Its associated eigenvalue, similar to the first eigenvector, is negative real, leading to rapid oscillatory linear dynamics and inactivation for rectified dynamics (Fig. 4i). Despite the similar dynamics, the synaptic weight matrix revealed complete mutual inhibition between R4d ring neurons (Fig. 4j, off-diagonal blue). This connectivity pattern has been identified in prior work[23], but its functional relevance remains incompletely understood.

One possible computation consistent with this connectivity pattern is a winner-take-all (WTA) computation with respect to visual features distributed across space. In a WTA circuit, the most strongly activated neuron strongly suppresses all other neurons, thereby preventing its own inhibition. To test this prediction, we simulated uniform visual drive to the R4d inhibitory sub-network while providing one neuron

with higher input (Fig. 4k, left and Methods). We found that the neuron with a stronger input indeed had a robust sustained response, whereas the other neurons' responses quickly decayed near to baseline (Fig. 4k). We confirmed that this WTA property is not specific to a single neuron by providing different neurons with biased input (Fig. 4k, right). Indeed, these dynamics persisted, which supports the idea that a WTA computation is a robust property of this circuit. On the basis of these findings, the R4d inhibitory sub-network appears well-poised to implement WTA dynamics and thus to select individual visual spatial channels as primary inputs to the central complex. We note that global mutual inhibition within ring neurons such as the R4d cell type has been characterized[24] and predicted to potentially drive WTA, but here we demonstrate that a mechanism predicted directly from anatomical parameters generates this computation.

We have analysed two of the hundreds of sparse eigenvectors revealed by an eigendecomposition of the fly connectome. We emphasize that these eigenvectors should not be interpreted as the 'true' effectome eigenvectors; rather, they provide a principled approach for generating and testing falsifiable hypotheses about causal relationships between neurons and the computations they may support. Our analyses served to demonstrate how neurons may be systematically chosen for causal perturbation experiments, and how—once the true effectome weights are learned for this subset of neurons—one might generate hypotheses about neural function than can be tested in vivo. Crucially, the dynamical mechanisms of the computations indicated by our simulations remain untested and require estimating the effectome between these neurons.

## Discussion

We developed a combined experimental and statistical approach to estimate a causal model of the fly central nervous system—its effectome. In simulation, we demonstrated that the approach provides consistent estimates of the ground truth effectome. We found that the huge number of parameters of a whole-brain model made this estimation unfeasible, motivating us to use the connectome as a prior to markedly increase the efficiency of our estimator. We analysed our connectome prior to reveal thousands of small putative circuits operating largely independently of each other. This indicates that whole-brain dynamics may be efficiently explained with sparse imaging and perturbation, which is far more feasible than dense imaging and perturbation. We analysed two of these circuits to find that one recapitulates a proposed circuit for computing spatial opponent motion, and the other provides an explicit mechanism for visual spatial selectivity in the ellipsoid body.

### Related IVs work

IVs has been an area of intense interest outside of the neurosciences but was only recently recognized as a useful tool for neural perturbation analysis[4,5,25,26]. Non-parametric approaches that estimate average functional effects[4,26] do not provide an explicit model of neural dynamics or differentiate between direct versus indirect synaptic effects. Yet, these non-parametric estimates are naturally more robust and could be used to validate model-based predictions. To our knowledge, our work is the first to provide a consistent estimator of a neural dynamics model in the challenging but nearly universal experimental conditions where there are potential unobserved confounders (for example, unobserved neurons).

An extension of this approach to higher order auto-regressive models would relax the restriction that the timescale of interaction is known and fixed (1 ms in our simulations); thus, slower effects (potentially through extra-synaptic paths such as peptide signalling pathways) could be detected. There has been recent progress in this direction[27], but it does not currently allow recurrent interaction between observed and unobserved populations, which is typical in neuroscience settings.

### Related connectomic work

There has been sustained interest in the analysis of large-scale anatomic information[10,28–30]. A linear systems analysis of the worm[30] made several findings that are qualitatively similar to ours: they 'rediscovered' several known circuits, found a preponderance of fast oscillatory and monotonic decaying modes, and identified sparse eigencircuits (see fig. 8 in ref. 30).

Graph-based fly connectome analyses[14,29] find that the fly brain is a small-world network with short paths between almost any pair of neurons, which could imply highly effective global communication. By contrast, we find that neural dynamics are best described by small, independent subsets of neurons: stimulating one eigencircuit has little effect on neurons outside of that circuit. Future perturbative work is required to determine the efficacy of global information propagation across the fly brain.

Along a similar line, a clustering algorithm has been applied to the fly hemibrain connectome[28,29]. It also recovered well-studied circuits, indicating its promise. Yet, the dynamics models under which the recovered circuits do in fact sustain distinct computations is unclear.

Anatomical information has been used to constrain mechanistic whole-brain models of the worm fit to neural data[31], and a similar model could be applied to the fly. A critical distinction of our approach is that it provides consistent estimates under unobserved confounders. Even for whole-brain recordings (which are now possible in the worm), where one might assume there are no unobserved variables, it is impossible to directly test this assumption. Yet this prior work directly demonstrates that connectomic constraints can improve the efficiency of model estimation—as we show in simulation.

In an alternative connectome-based approach applied to the visual system of the fly[32], parameters were not constrained to neural activity but were instead optimized to perform discrimination of visual motion. By sharing parameters across circuits that are thought to perform similar functions, the efficiency of estimation was greatly increased. In a similar approach, IV–Bayes could be extended to a hierarchical Bayesian model that borrows statistical power across circuits and cell types that are hypothesized to have similar functional properties.

We were able to recapitulate experiments on the visual system by directly stimulating model neurons in the central nervous system (Fig. 4), but this required prior knowledge of how specific sets of stimuli affect the central nervous system. Incorporating the effects of stimuli on early sensory neurons (for example, ommatidia) is a critical direction for utilizing effectome estimates to predict sensory computations for novel stimuli (for example, in refs. 32,33). Linking estimated internal dynamics models to behaviour—a central goal of systems neuroscience—will require integration with models of how descending motor neurons actuate the body[34–36].

### A broader definition of the effectome

The method that we introduce here is a consistent estimator of a linear dynamical system, yet the fly nervous system is a highly nonlinear system. To provide a concrete interpretation of linear model estimates applied to nonlinear neural dynamics, we analysed our estimator in the context of a conductance-based model of neural dynamics. This model includes both a spiking nonlinearity and nonlinear synaptic integration. We find that the IV estimate converges to the Jacobian (matrix of partial derivatives with respect to voltage of each neuron) of the underlying neural dynamics equation, evaluated at the voltages of the neurons in the population (Extended Data Fig. 3 and Supplementary Information, 'Conductance-based neural dynamics model'). This result is entirely consistent with our notion of the effectome—the Jacobian captures the effect of a small perturbation of one neuron on any other neuron. Our linear simulation (equation (1)) is a special case in which the Jacobian is exactly the connectome. In the conductance model, the effect of a perturbation varies with the state of the

network and the Jacobian captures this notion because it varies with respect to neuronal voltage. This also provides motivation to titrate optogenetic stimulation to the minimal amount possible—for larger deviations, the estimate of the Jacobian becomes less precise and averages over more states. The Jacobian is also a foundational quantity in the study of stochastic differential equations. It is classically utilized in a linear dynamical system that approximates nonlinear and non-stationary dynamics along an estimated trajectory. For example, in the case of neural data, independent IV estimates can be formed across a stimulus-triggered average. More generally, extensions to our estimator can utilize estimated trajectories within repeated experimental conditions, stereotyped behaviour or inferred latent states[37]. The latter may be critical if the emerging observation that the bulk of variation in global brain dynamics is unrelated to stimuli or behaviour holds in the fly[38–40]. Ultimately, we interpret effectome estimates as capturing local interactions between neurons in a particular state (for example, at a set of membrane voltages). Further research is needed to understand how to use these local estimates to learn about changes in state (such as synaptic plasticity).

Our interpretation of our estimator as providing the Jacobian of neural dynamics further clarifies the utility of the connectome prior. We show analytically in our conductance model that the Jacobian between unconnected neurons will always be zero (Supplementary Information, equation 6). Thus, the connectome prior mean will be wrong for only 0.01% of the effectome parameters, because only 0.01% of pairs of neurons form a synapse in the fly brain. Thus, even with misspecified non-zero weights IV–Bayes provides a large gain in statistical efficiency (Extended Data Fig. 4). In general, most neurons in the fly brain do not directly affect each other, and this is where the majority of our gains in statistical efficiency are achieved. This in combination with weak priors on the small subset of neurons that plausibly directly affect each other allows our estimator the flexibility to efficiently estimate the effectome. We also confirm that even with a completely misspecified prior (for example, there may be extra-synaptic effects through peptide signalling pathways[41]), our estimator, with enough data, converges to the Jacobian—that is, the weight matrix that provides a linear description of the effect of each neuron on any other in the system's current state (Extended Data Fig. 5).

The connection of the effectome to the Jacobian of conductance model also provides a concrete interpretation of the connectome (matrix of signed synaptic counts) as a linear dynamics matrix. It is in only a narrow set of situations that the best linear approximation to these nonlinear neural dynamics would be proportional to the connectome. Variation in neural voltage across neurons, synaptic reversal potentials, synaptic conductance, membrane time constants and more can all corrupt proportionality (Supplementary Information, equation 6). Thus, a biophysical interpretation of our results is that the eigencircuits decompose the best linear approximation to neural dynamics under the assumption that there is a small amount of variation across these neuronal properties. A mild confirmation of this untested assumption is that we do in fact recover known functional sub-circuits of the fly nervous system from an eigendecomposition of the connectome (Fig. 4).

In general, the form of dynamics in the fly brain even for our simplified conductance model is highly under-constrained—the parameters needed to evaluate the Jacobian are not available (including average voltage, membrane time constants, postsynaptic currents associated with different synapse types and morphologies). It is beyond the scope of this work to identify biophysically realistic models of neural dynamics from the connectome and measured neural activity. Our estimate of the effectome represents a first-order approximation to these underlying dynamics, which will depend on the state of the nervous system. Future work could focus on how sets of these first-order estimates can be used to infer the appropriate models and parameters of nonlinear neural dynamics equations.

## Sufficient experimental setting

Here we specify the sufficient, feasible, experimental setting to 'learn' the effectome. We mean feasible in that the technologies to perform these experiments exist but have yet to be simultaneously integrated into the fly. In Supplementary Information, 'Experimental approach', we discuss how to achieve this ideal setting, including practical steps for generating a fly line specific to estimating an eigencircuit.

We note that these requirements are sufficient but that there could certainly be alternative approaches to satisfying the requirements of our estimator. In general, we expect that it would be unfeasible to estimate a large fraction of the effectome in a single experiment. Thus, our proposed strategy is to estimate eigenvectors of the effectome piecemeal, with sparse subsets across genetically identical flies in the same experimental conditions (Supplementary Information, 'Experimental approach').

There are four main technical requirements to estimate the effectome in such a manner. (1) The ability to select a sparse subset of neurons within a population of interest to image and stimulate (Supplementary Information, 'Sparse expression'). (2) The ability to record the intracellular voltage of a population of neurons (Supplementary Information, 'Voltage imaging'). (3) Simultaneous independent direct stimulation of neurons in the population (Supplementary Information, 'Holographic stimulation'). Both the recording and stimulation must be at the timescale of neuronal interaction. (4) Identification of imaged and stimulated cells with the connectome (Supplementary Information, 'Identification pipeline').

This ideal experimental setting is challenging to achieve. Nonetheless, if it is achieved, it provides a principled approach to accurately estimate a causal account of local neuronal dynamics. A complete nonlinear dynamical account of brain dynamics would necessarily recapitulate local dynamics; thus, the effectome will serve to rigorously constrain and test a mechanistic model of the nervous system.

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

## Methods

### Statistical model of fly brain

Here we approximate fly brain dynamics as a first-order vector auto-regressive model (VAR(1)),

$$\mathbf{r}_t = W\mathbf{r}_{t-1} + W_{l,x}L_t + \epsilon_t. \tag{3}$$

where $\mathbf{r}_t$ is the $D \times 1$ vector of responses from all neurons on time step $t$, $W$ is the full $D \times D$ dynamics matrix, $L_t$ is the random $D_l \times 1$ vector of stimulation power (for example, voltage), $W_{l,x}$ is the matrix specifying the effect of stimulation, and $\epsilon_t$ is additive noise with arbitrary bounded covariance (for graphical model, see Extended Data Fig. 1). The raw IV estimator that we outline below requires no distributional assumptions on the random variables ($L_t$, $\epsilon_t$). Furthermore, the noise ($\epsilon_t$) can take on any correlation structure over time and across neurons, so our estimator applied to dynamics of this form will converge to the true effectome $W$, regardless of confounding inputs (so long as the noise is independent of the stimulation $L_t$). We note that for nonlinear dynamics, further assumptions are required (see Supplementary Information, 'IV estimator applied to nonlinear dynamics').

For our derivations below we define three subsets of neurons in $\mathbf{r}_t$, we will call the vector of neurons that are observed in a given experiment, target neurons, at time point $t$, $Y_t$, the neurons that are both stimulated by the laser and observed $X_t$, source neurons, and finally neurons that are neither observed nor stimulated $Z_t$ (for graphical model, see Extended Data Fig. 1). Note that source neurons are a subset of target neurons.

### Simulations

**Simulations to analyse eigencircuits.** To evaluate neural dynamics associated with eigencircuits (see 'Dominant circuits are interpretable') in continuous time, we restricted our analyses to neurons with the highest eigenvector loadings that accounted for 75% of all eigenvector power. In the case of the opponent motion circuit, this was 5 neurons, and in the case of the ellipsoid body circuit, this was 21 neurons. We set the discrete time step to $T = 10$ ms, and a sampling period of $\Delta = 1$ ms so that at each time step,

$$\mathbf{r}_{t+\Delta} = \exp(\log(W)\Delta T^{-1})f(\mathbf{r}_t + \mathbf{u}_t) \tag{4}$$

where $W$ is the putative effectome of the subset of neurons being simulated, $f(\cdot)$ is a nonlinearity (here rectification), and $\mathbf{u}_t$ is a vector of time-varying inputs.

We tuned the membrane time constant manually to 0.5 ms to recapitulate the qualitative findings of prior work. The resulting $W$ was scaled to have a maximum eigenvalue of 1. It is unknown what a good approximate scaling of synapse number to linear effect is in these regions but if the scaling was zero then this circuit would not perform a computation and if it was too large the dynamics would be unstable. The membrane time constant is often faster in sensory systems but depends on the voltage of the neuron[42]. Future work could consider other normative motivations for scaling weights and time constants (for example, ref. 32).

In both the ellipsoid and opponent motion simulations, inputs in Fig. 4 were simulated by adding in a step function input to the relevant neurons in our dynamics simulation. This step function was either at 0 or a hand tuned maximum. In the opponent motion circuit input maximum was 0.01 and for the ellipsoid body circuit the winner neuron receiving a larger input had input of 0.01 and other neurons had 0.006. In both cases stimulation lasted 150 ms. In the case of the opponent motion circuit, we used prior literature to determine which neurons to stimulate under unilateral and bilateral BTF motion[20,21]. In the case of the ellipsoid body circuit, the ordering of the neurons with respect to retinotopic input was arbitrarily set by their eigenvector loading magnitude.

**Simulations to evaluate estimators.** In all simulations to evaluate statistical estimators, $\epsilon_t \sim N_D(0, \Sigma_\epsilon)$ where $\Sigma_\epsilon = cI_D$ and $L_t \sim N_D(0, lI_D)$. To simulate misspecification of the connectome prior mean, we estimated the accuracy of our estimator across many 'ground truth' effectomes drawn from the connectome prior (except without a small constant added to the variance so that synaptic weights equal to 0 remained 0), such that the connectome prior mean was never the same as the 'ground truth' effectome in a given simulation.

Given that there are a host of unknowns with respect to a real experimental setting (such as imaging SNR, strength of laser effect or duration of recordings), we hand tuned these parameters to give reasonable rates of convergence. In our whole-brain IV simulations, signal-to-noise ratio (SNR) = 10 (laser power relative to noise power). We note that while we have simulated from a parameter regime in which our estimator converges rapidly, there are many parameter settings where convergence is slow. We note that conditions of high noise and little effect of the laser are particularly challenging. Slow timescales are also challenging because they effectively filter out most of the power of the white noise perturbation (but for extension to correlated IVs, see ref. 27). Yet, we show analytically that our estimator is consistent and thus will converge with enough samples.

For clarity, in our example simulation we chose a single neuron to stimulate and estimate its downstream synaptic weights (Fig. 3). This neuron was chosen because it had a larger than typical number of downstream contacts. It is straightforward to estimate downstream weights for multiple neurons simultaneously (equation (7)), and we demonstrate this both for IV and IV–Bayes (Extended Data Figs. 4–6) and IV in a conductance-based model (Extended Data Figs. 3 and 4). We note that in our simulations, as the number of independent perturbations increases ($n_l$), there is effectively more noise overall both through second-order effects and in estimating the effect on a downstream neuron that itself is being perturbed. This is another pressure to keep the number of stimulated neurons low and the strength of perturbation minimal, but depends on the particularity of connectivity.

### IV estimator for an LDS

We note that

$$\text{Cov}[X_t, L_t] = W_{l,x} \tag{5}$$

because the stimulation is assumed to have identity covariance. Thus, by calculating the sample covariance between the laser and simultaneous activity in the stimulated neurons we can obtain an unbiased estimate of the linear weighting of laser drive on each neuron. Similarly, we can obtain an unbiased estimate of the linear effect of the laser on all target neurons at the next time step,

$$\text{Cov}[Y_{t+1}, L_t] = W_{x,y}W_{l,x}, \tag{6}$$

where $W_{x,y}$ is the submatrix of $W$ with postsynaptic effects of $X_t$ on $Y_{t+1}$. We can then use equation (5) to identify $W_{x,y}$ with

$$W_{x,y} = W_{x,y}W_{l,x}(W_{l,x})^+, \tag{7}$$

where $(W_{l,x})^+$ is a pseudo inverse because we have not specified the rank of $L$. Only if $n_L \geq n_s$ is this a true inverse and $W_{l,x}$ is invertible. An equivalent but more intuitive approach is termed two-stage least-squares (2SLS), where in the first stage $L_t$ is regressed on $X_t$ to give $\hat{X}_t = \hat{W}_{l,x}L_t$ and then $\hat{X}_t$ is regressed on $Y_{t+1}$ to give $\hat{Y}_{t+1} = \hat{W}_{x,y}\hat{X}_t$. The IV estimator can also be extended to higher order AR processes by conditioning the estimator on multiple past time steps[27].

We note that multi-synaptic effects can be derived from the estimated monosynaptic effects with powers of the effectome matrix. For example, if we have the effectome matrix $W$ and input $r$ is an input vector of all zeros except for neuron $i$, then the $n$th order synaptic effect is exactly

$W^n r$ (for example, $n = 1$ gives direct synaptic effects, $n = 2$ gives effects through up to two synapses, and so on).

## The connectome prior

We use the 2SLS approach to motivate a consistent estimator from a Bayesian perspective. In short, we perform classical Bayesian regression for the second stage of regression using the connectome as a prior on the weights $W_{x,y}$. To be consistent with the most typical setting of Bayesian regression we first work out the case of multiple source neurons and a single target neuron (that is, learning a set of weights in the same row of $W$). We assume the conditional distribution of the output given the input is:

$$y_t | \mathbf{x}_t \sim \mathcal{N}(\mathbf{w}^\mathsf{T} \mathbf{x}_t, \sigma^2), \tag{8}$$

where $(\mathbf{x}_t, \mathbf{y}_t)$ represents the input and output for sample $t \in \{1, \dots, T\}$, and $\sigma^2$ is the variance of the observation noise in $\mathbf{y}$.

Let us now suppose that $\mu$ provides the mean for a Gaussian prior over the linear weights $\mathbf{w}$:

$$\mathbf{w} \sim \mathcal{N}(\mu, \gamma^2 I). \tag{9}$$

Let $\mu = sc$, where the hyperparameter $s$ scales $c$, the connectome prior we set to be equal to the synaptic count and sign. Combining this prior with the likelihood defined above gives us the following posterior mean:

$$\begin{aligned}
\hat{\mathbf{w}}_{\mathrm{MAP}} &= \arg \max_{\mathbf{w}} P(\mathbf{w}|X, Y, \theta) \\
&= (1/\sigma^2 X^\mathsf{T} X + 1/\gamma^2 I)^{-1} (1/\sigma^2 X^\mathsf{T} Y + 1/\gamma^2 \mu) \\
&= (X^\mathsf{T} X + \sigma^2/\gamma^2 I)^{-1} X^\mathsf{T} Y + (\gamma^2/\sigma^2 X^\mathsf{T} X + I)^{-1} \mu,
\end{aligned} \tag{10}$$

where $\theta = \{\sigma^2, \gamma^2, c\}$ denotes the hyperparameters. The second expression above (equation (10)) shows that the maximum a posteriori (MAP) estimate is the standard 'ridge regression' estimate, $\hat{\mathbf{w}}_{\mathrm{ridge}} = \left( X^\mathsf{T} X + \frac{\sigma^2}{\gamma^2} I \right)^{-1} X^\mathsf{T} Y$, plus a term that biases the estimate towards the anatomical connectome $\mu$. Note that in the limit of small observation noise $\sigma^2$ or large prior variance $\gamma^2$, the MAP estimate converges to the maximum likelihood (ML) estimate, whereas in the limit of large $\sigma^2$ or small $\gamma^2$, it converges to $\mu$.

In our simulations we choose the optimal hyperparameters beforehand but the hyperparameters could be learned via a standard cross-validation grid search. A more principled approach would be to use evidence optimization,

$$\hat{\theta} = \arg \max_\theta P(Y|X, \theta) = \arg \max_\theta \int P(Y|X, \mathbf{w}, \theta) P(\mathbf{w}|\theta) d\mathbf{w}, \tag{11}$$

which would be straightforward given that the evidence is available in closed form for this model.

## Construction of fly connectome matrix

The connectome is a reconstruction of the central nervous system of a seven-day-old adult female *Drosophilia melanogaster*. We use the most recent version of the connectome v783. Details of the reconstruction are provided in the original publications of the connectome dataset[1].

Each entry in the connectome matrix $W$, the main object of study in our analyses, was the number of synapses multiplied by their inferred sign based on predicted neurotransmitter type[11]. Specifically, neurons with neurotransmitters acetylcholine and dopamine had positive weights on their downstream neurons and neurons with GABA, serotonin, glutamate and octopamine had negative weights. The neurotransmitter type was predicted directly from electron microscopy images trained on synapses with known neurotransmitter types. The matrix $W$ scaled for stability was used as the connectome prior mean in estimator simulations (Fig. 2) and our eigendecomposition analysis (equation (2); Figs. 3 and 4). A threshold was set on the synapse count such that any connections with less than five synapses were set to zero. This choice followed the reasoning of other analyses of the connectome[10] that this would minimize the impact of spurious synapses—manual proofreading did not extend to connections with fewer than five synapses.

## Reporting summary

Further information on research design is available in the Nature Portfolio Reporting Summary linked to this article.

## Data availability

Data used in this study can be downloaded from https://codex.flywire.ai/api/download. FlyWire data can be accessed online through Codex (Connectome Data Explorer) at https://codex.flywire.ai. Codex provides neuron annotations, neurotransmitter information, and compact data downloads. All eigencircuits' loadings and additional plots of eigencircuits are available at https://github.com/dp4846/conn2eff.

## Code availability

All code to analyse data and generate figures is available at https://github.com/dp4846/conn2eff.

42. Koch, C. *Biophysics of Computation: Information Processing in Single Neurons* (Oxford Univ. Press, 2004).

**Acknowledgements** The authors thank A. Lin for assistance with generating neuron renders; O. Tambour, R. Pang and M. Creamer for insightful and inspiring discussions; the Princeton FlyWire team and members of the Murthy and Seung laboratories, as well as members of the Allen Institute for Brain Science, for development and maintenance of FlyWire (supported by BRAIN Initiative grants MH117815 and NS126935 to M.M. and S. Seung); J. Funke and his group for contributing neurotransmitter IDs; and members of the Princeton FlyWire team, the Cambridge Drosophila Connectomics group led by G.S.X.E.J. and the FlyWire consortium for neuron proofreading and annotation. J.W.P. was supported by grants from the Simons Collaboration on the Global Brain (SCGB AWD543027) and the NIH BRAIN initiative (NS104899 and 9R01DA056404-04). D.A.P. was supported by Simons Collaboration on the Global Brain (SCGB AWD543027). M.J.A. was supported by NIH NINDS R35 1R35NS111580-01 and BRAIN NINDS R01 NS104899. G.S.X.E.J. was supported by Wellcome Trust Collaborative Awards (203261/Z/16/Z and 220343/Z/20/Z) and a Neuronex2 Award (MRC MC_EX_MR/T046279/1), and received core support from the MRC (MC-U105188491). M.M. acknowledges support from the National Institutes of Health (NIH) BRAIN Initiative RF1 MH117815, RF1 MH129268 and U24NS126935 and from the Princeton Neuroscience Institute, as well as assistance from Google.

**Author contributions** D.A.P. conceived of the presented idea. D.A.P. and M.J.A. analysed the data and ran simulations. D.A.P. and J.W.P. derived frequentist and Bayesian components of the estimator, respectively. D.A.P., M.J.A. and J.W.P. wrote the manuscript with feedback from M.M. J.W.P. supervised the project. S.D., P.S. and A.M. curated the data and made it available for download. A.M. and A.R.S. built the Codex online platform. Sc.Y., C.E.M., M.C., K.E. and P.S. trained and managed Flywire proofreaders. G.S.X.E.J. provided neurotransmitter information. C.E.M. and A.R.S. provided Flywire community support and training. M.M. led the FlyWire project and G.S.X.E.J. led the Cambridge proofreading.

**Competing interests** The authors declare no competing interests.

**Additional information**
**Correspondence and requests for materials** should be addressed to Dean A. Pospisil or Max J. Aragon.

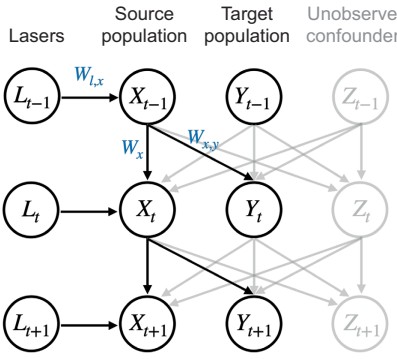

**Extended Data Fig. 1 | Graphical model associated with VAR(1) estimation.**
$L_t$ is IID stimulation at time step t, effect on source population is immediate
and mediated by linear transformation $W_{l,x}$. Effect of $X_{t-1}$ on $X_t$ and $Y_t$ (target,
unstimulated population) is mediated respectively by linear transformation
$W_x$ and $W_{x,y}$. Interacting unobserved confounds can add arbitrary correlated
noise ($Z_t$).

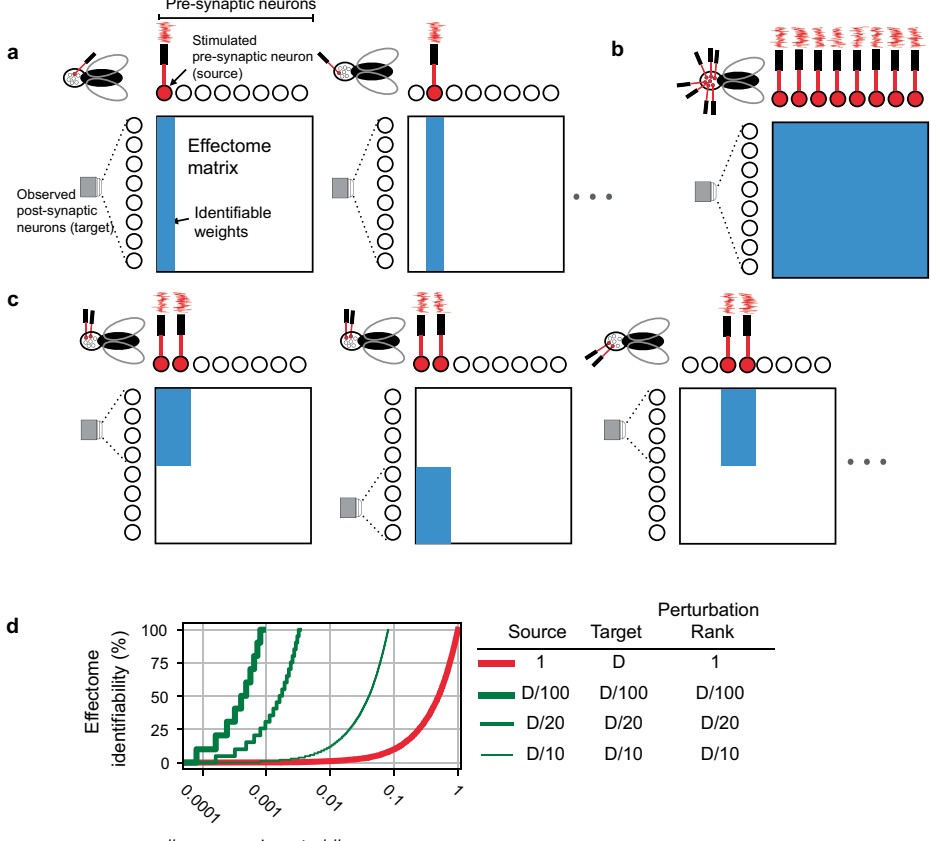

**Extended Data Fig. 2 | Schematic of perturbation experiment types.**
**(a) (left)** Single fly experiment with whole-brain observation (microscope FOV encompasses all postsynaptic neurons), single neuron stimulation (laser drives filled red neuron), and total rank of stimulation is 1 (only one laser). As a result, all postsynaptic weights from the stimulated neuron are identifiable (blue column of weight matrix filled). **(middle)** To learn all other effectome weights in this setup would require as many flies as neurons, as each individual neuron is stimulated. **(b)** Single fly experiment in which all weights are identifiable: whole-brain observation, whole-brain stimulation, and rank of stimulation is equal to the number of neurons in the brain (same number of lasers as neurons). **(c) (left)** Single fly experiment with partial brain observation (FOV encompasses

half of neurons), partial brain stimulation (two neurons red filled), and rank of stimulation is equal to that of the source (2). **(middle)** To identify all weights of source neurons the experiment is repeated in another fly but with different target neurons. All effectome weights can be identified piecemeal in this manner. **(d)** Convergence rate to full identifiability of the fly effectome as a function of the fraction of number of experiments over the total number of neurons. Different traces reflect different experimental settings. Columns in legend are three primary ways experiments can vary. Source is the number of neurons that are being stimulated. Target is the number of neurons observed, we assume the source neurons are also observed. Perturbation rank is the dimensionality of the perturbation method. $D$ is the total number of neurons.

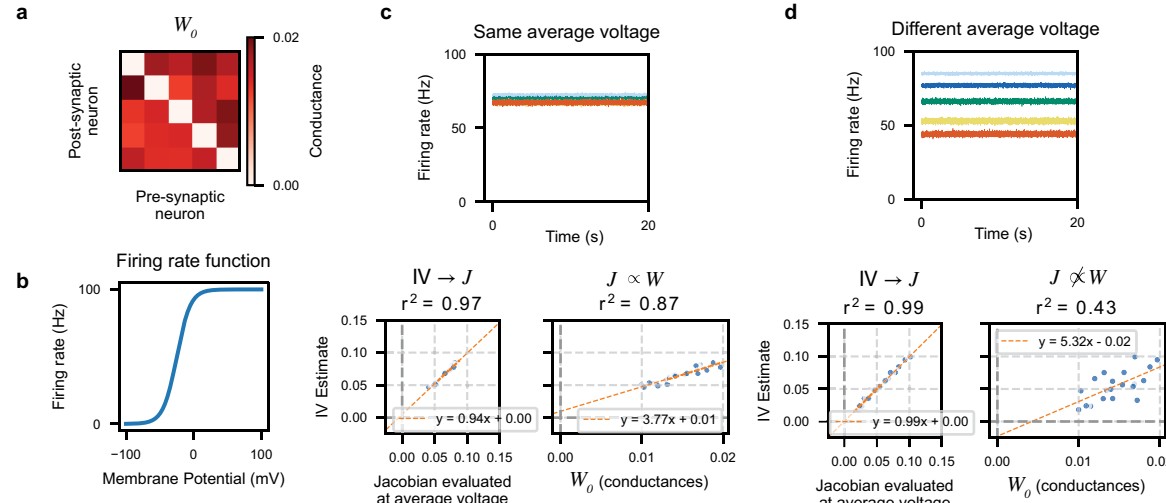

**Extended Data Fig. 3 | Instrumental variables estimator converges to the Jacobian of membrane voltage equation. (a)** Conductance matrix, $W_0$, of conductance simulation where off-diagonals were set randomly according to a uniform distribution between 0.01 and 0.02. The reversal potential of all synapses is 0 mV; thus all synapses are typically excitatory. Membrane constant $\tau = 10$ ms, membrane resistance $R = 1$, input noise is SD = 2 mV, laser perturbation SD = 2 mV, inputs were hand chosen to induce two conditions of similar vs different average voltages, and the conductance equation is integrated using Euler's method with $\Delta = 1$ ms with noise and laser perturbation added at each time step. **(b)** The firing rate as a function of membrane voltage is sigmoidal. **(c) (top)** Example traces of firing rate over time for neurons with

similar average voltage. Different colors represent the five different neurons. **(c) (bottom left)** Relationship between IV estimates and the Jacobian evaluated at the average voltage, with a high correlation ($r^2 = 0.97$) and a linear fit with slope near 1 indicating IV converges to the Jacobian of neural dynamics. **(c) (bottom right)** Relationship between the Jacobian and conductances ($W_0$), with a strong correlation ($r^2 = 0.87$) and slope deviating from one indicating that approximately $J \propto W$. **(d) (top)** Example traces of firing rate over time for neurons with different average voltages. **(d) (bottom left)** IV estimates converge to the Jacobian again. **(d) (bottom right)** In contrast, Jacobian and conductance matrix are not proportional to each other when average voltages vary widely.

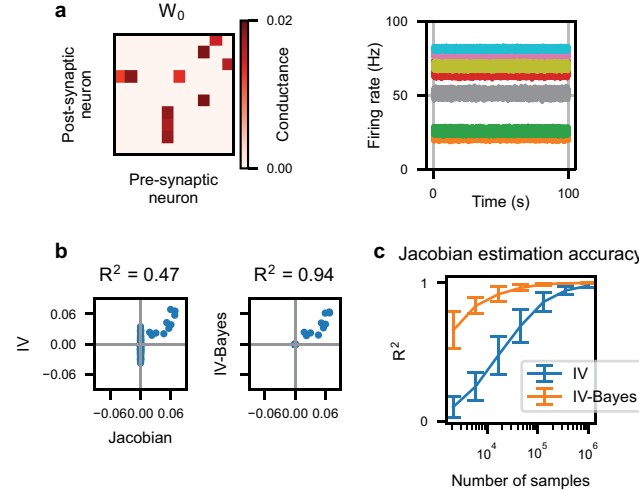

**Extended Data Fig. 4 | IV-Bayes improves the efficiency of Jacobian estimation in the conductance model. (a) (left)** Example of sparse synaptic connectivity matrix $W_0$ used for conductance simulations, where off-diagonals were set randomly according to a uniform distribution between 0.01 and 0.02, then on average 90% of these were set to 0. **(a) (right)** Example firing rate traces across 10 neurons. **(b)** Comparison of IV and IV-Bayes estimate of Jacobian. **(c)** Comparison of $R^2$ values for IV and IV-Bayes estimators as a function of the number of samples. The plot shows that the IV-Bayes estimator achieves higher average $R^2$ values with fewer samples compared to the IV estimator. Error bars show standard deviations across 5 simulations (a different $W_0$ chosen for each simulation but all other parameters remain the same).

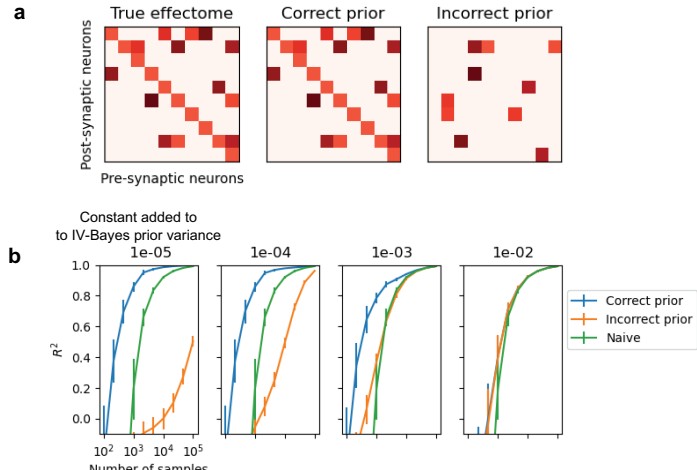

**Extended Data Fig. 5 | IV-Bayes is a consistent estimator even with incorrect prior. (a) (left)** Example ground truth simulated effectome for linear simulations (off-diagonal weights drawn from uniform distribution ([0.1,0.2]), then on average 9- % are set to 0 and diagonal is set to 0.1) **(a) (center)** In 'Correct prior' condition prior mean is set to the true effectome. **(a) (right)** In 'Incorrect prior' condition prior mean is set to independent effectome (without diagonal).

**(b)** Simulations of IV-Bayes estimator with correct (blue) and incorrect prior (orange) and the raw IV estimator (green) across number of samples (i.e., duration of recording) and resamplings of effectomes (mean ± s.d.; n = 10 simulations). Variance of prior mean has a constant (see Fig. 2c legend) of increasing size added to it (increasing left to right). $R^2$ is measured between the estimate of the effectome and ground truth .

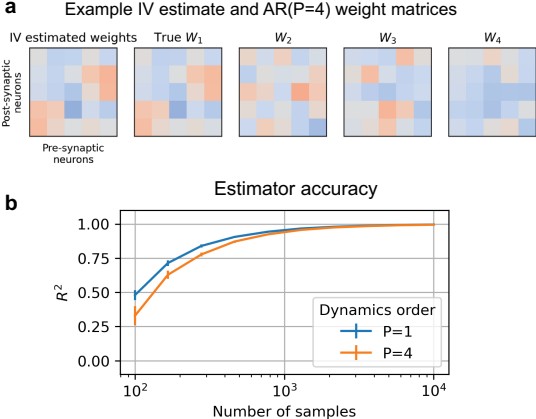

**a**  Example IV estimate and AR(P=4) weight matrices

IV estimated weights    True $W_1$    $W_2$    $W_3$    $W_4$

Post-synaptic neurons

Pre-synaptic neurons

**b**  Estimator accuracy

$R^2$

Dynamics order
— P=1
— P=4

$10^2$    $10^3$    $10^4$
Number of samples

**Extended Data Fig. 6 | IV converges to first order effects in a higher order AR model. (a) (left)** Example VAR(1) IV estimate from data drawn from VAR(4). **(a) (right)** Example VAR(4) process weights drawn from an IID standard normal then scaled for stability. Note estimate and $W_1$ are nearly identical. **(b)** Accuracy of recovery of the first order effects ($W_1$) as a function of the number of time samples (mean ± s.d.; n = 50 simulations).

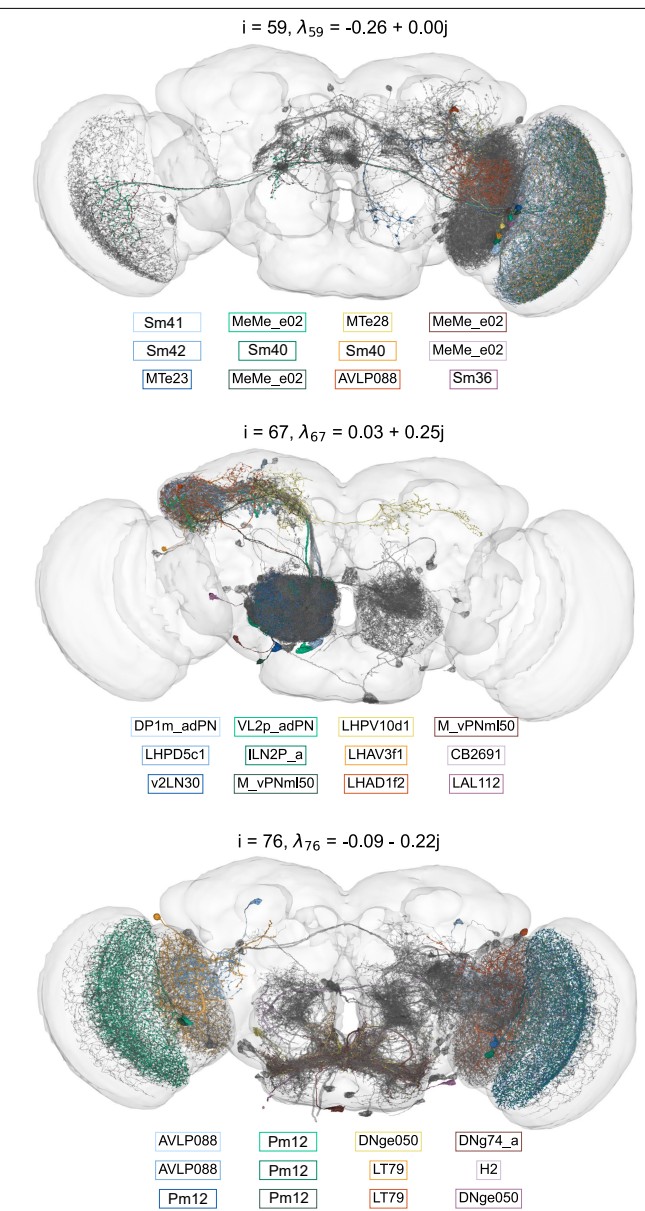

i = 59, $\lambda_{59}$ = -0.26 + 0.00j

| | | | |
|---|---|---|---|
| Sm41 | MeMe_e02 | MTe28 | MeMe_e02 |
| Sm42 | Sm40 | Sm40 | MeMe_e02 |
| MTe23 | MeMe_e02 | AVLP088 | Sm36 |

i = 67, $\lambda_{67}$ = 0.03 + 0.25j

| | | | |
|---|---|---|---|
| DP1m_adPN | VL2p_adPN | LHPV10d1 | M_vPNml50 |
| LHPD5c1 | ILN2P_a | LHAV3f1 | CB2691 |
| v2LN30 | M_vPNml50 | LHAD1f2 | LAL112 |

i = 76, $\lambda_{76}$ = -0.09 - 0.22j

| | | | |
|---|---|---|---|
| AVLP088 | Pm12 | DNge050 | DNg74_a |
| AVLP088 | Pm12 | LT79 | H2 |
| Pm12 | Pm12 | LT79 | DNge050 |

**Extended Data Fig. 7 | Examples of non-localized eigencircuits.** Top: Circuit with neurons in the optic lobes and central complex. Middle: Circuit with neurons in the antennal lobes, lateral horn, and lateral accessory lobe. Bottom: Circuit with neurons in the optic lobe and pre-motor regions.

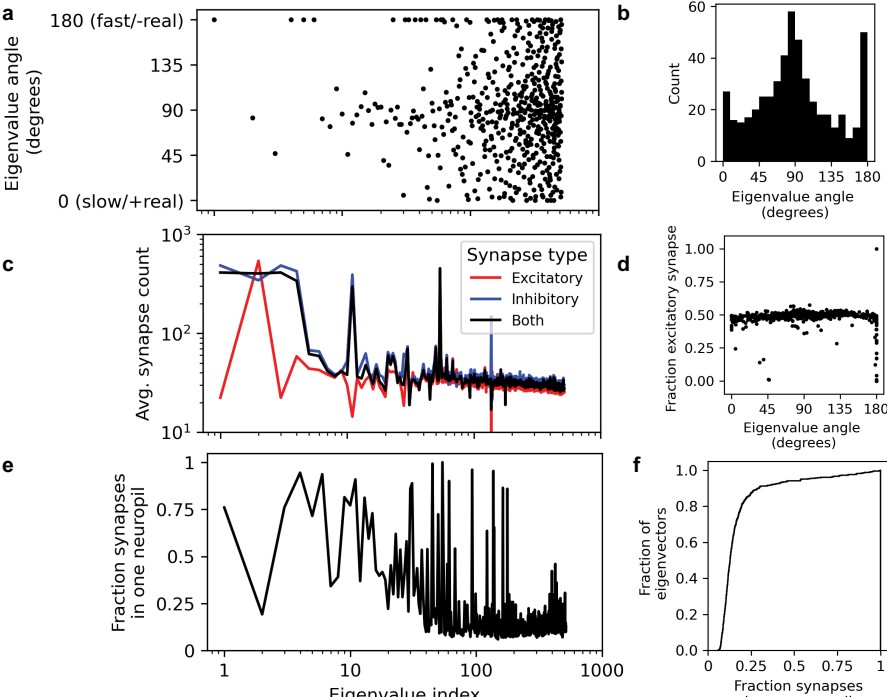

**Extended Data Fig. 8 | Summary statistics of time scales, locality, and synapse properties. (a)** Angle in complex plane of eigenvalues associated with eigencircuits plotted as a function of eigencircuit rank (w.r.t. eigenvalue magnitude). The angle 0 implies the slowest possible dynamics where eigencircuit monotonically decays. The angle 180 implies the fastest possible dynamics where the sign of the eigencircuit flips at every discrete time step. **(b)** Histogram of angles. **(c)** Average synapse count of excitatory neurons that form synapses in the top 75% of eigenvector loading (red), average for inhibitory (blue), and across both (black). **(d)** Plot of the fraction of excitatory synapses of all synapses formed in the top 75% of eigenvector loading of each eigencircuit as a function of time scale (eigenvalue angle). **(e)** Locality index, fraction of synapses in one neuropil for the top 75% of eigenvector loadings of eigenvector loading of each eigencircuit. **(f)** Cumulative distribution of (E).

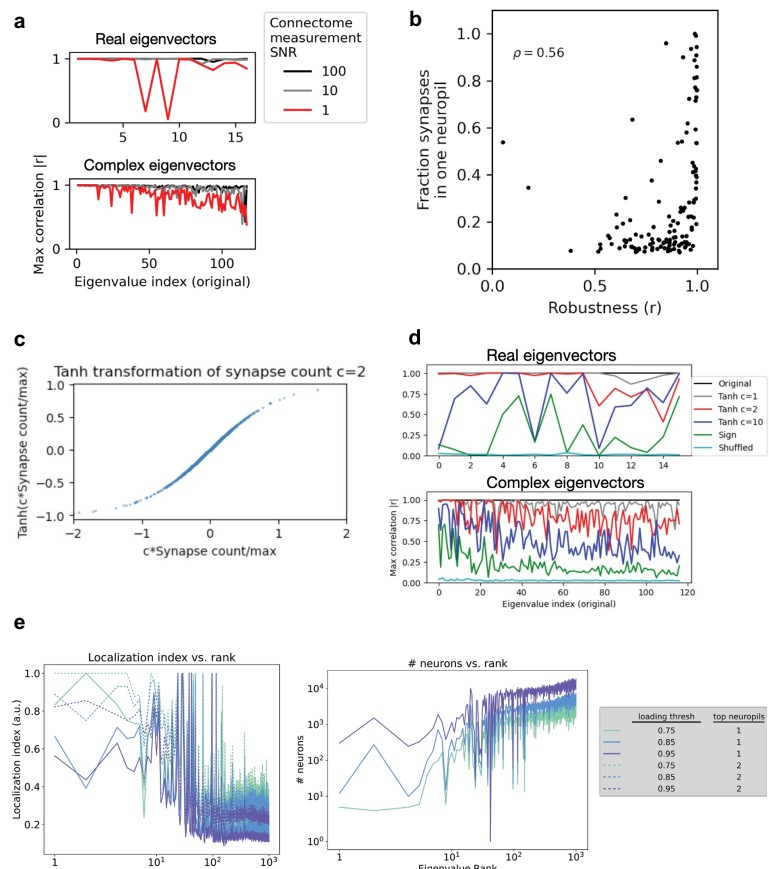

**Extended Data Fig. 9 | Robustness of eigencircuits. (a)** Robustness to measurement error. Maximum absolute correlation between each original eigencircuit and all eigencircuits from the connectome with Gaussian noise added. For complex eigenvectors, we regressed the real and imaginary components onto those of another then calculated the r-value of this two-parameter linear fit. Measure was taken for three levels of noise (SNR = 100 $\sigma^2 = 0.01\mu$ black, SNR = 10 $\sigma^2 = 0.1\mu$ grey, and SNR = 1 $\sigma^2 = \mu$ red where $\mu$ is the synapse count). **(b)** Non-anatomically localized eigencircuits tend to be less robust to measurement error. Robustness of eigencircuits measured as maximum absolute correlation with eigencircuits from noisy connectome

(SNR = 1, see Extended Data Fig. 9 red trace) plotted against locality index, fraction of synapses in one neuropil for top 75% of eigenvector loading of each eigencircuit. **(c)** Robustness of eigencircuits to hyperbolic tangent transformation (tanh) of connectome weights scaled by half the max synapse count (c = 2). **(d)** Plot of maximum absolute correlation of original vs transformed connectomes (see A) versus rank of original eigencircuit, split across real and complex unique eigencircuits of top 250 eigencircuits. **(e)** Robustness of non-localized eigencircuits to choices in loading threshold. **(left)** Localization index computed across different inclusion criteria. **(right)** Number of neurons within each eigencircuit for multiple loading thresholds.

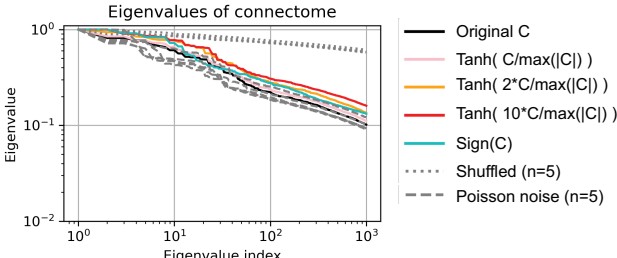

**Extended Data Fig. 10 | Effect of nonlinearity and measurement error applied to connectome on dimensionality.** For reference, the original eigenvalues of the connectome (scaled by the largest magnitude eigenvalue) are plotted (black). Non-linearity of increasing degree via hyperbolic tangent applied to connectome weights scaled relative to their maximum (larger scaling, stronger effect of nonlinearity) shows a small increase in dimensionality (pink, orange, red above black). In an extremal case, the connectome weights were set to their sign so that all entries were either +1, 0, or −1, and dimensionality was similar (cyan overlaps black). A shuffle control, where the index of count entries was shuffled without replacement, was used to determine if dimensionality was the result of marginal connectome statistics (e.g., sparsity). This showed a large increase in dimensionality (n = 5 independent shuffles, dotted lines well above black). Simulated measurement error was added with draws from Poisson distributions with means equal to the original synapse count and sign was drawn from a binomial distribution with probability set by the neurotransmitter type prediction confidence (n = 5 simulations, dashed line).

# Reporting Summary

## Statistics

For all statistical analyses, confirm that the following items are present in the figure legend, table legend, main text, or Methods section.

| n/a | Confirmed | |
|---|---|---|
| ☒ | ☐ | The exact sample size (*n*) for each experimental group/condition, given as a discrete number and unit of measurement |
| ☒ | ☐ | A statement on whether measurements were taken from distinct samples or whether the same sample was measured repeatedly |
| ☒ | ☐ | The statistical test(s) used AND whether they are one- or two-sided<br>*Only common tests should be described solely by name; describe more complex techniques in the Methods section.* |
| ☒ | ☐ | A description of all covariates tested |
| ☒ | ☐ | A description of any assumptions or corrections, such as tests of normality and adjustment for multiple comparisons |
| ☒ | ☐ | A full description of the statistical parameters including central tendency (e.g. means) or other basic estimates (e.g. regression coefficient) AND variation (e.g. standard deviation) or associated estimates of uncertainty (e.g. confidence intervals) |
| ☒ | ☐ | For null hypothesis testing, the test statistic (e.g. *F*, *t*, *r*) with confidence intervals, effect sizes, degrees of freedom and *P* value noted<br>*Give P values as exact values whenever suitable.* |
| ☒ | ☐ | For Bayesian analysis, information on the choice of priors and Markov chain Monte Carlo settings |
| ☒ | ☐ | For hierarchical and complex designs, identification of the appropriate level for tests and full reporting of outcomes |
| ☒ | ☐ | Estimates of effect sizes (e.g. Cohen's *d*, Pearson's *r*), indicating how they were calculated |

*Our web collection on statistics for biologists contains articles on many of the points above.*

## Software and code

Policy information about availability of computer code

| Data collection | The connectome is a reconstruction of the central nervous system of a 7 day old adult female Drosophilia melanogaster. We use the most recent version of the connectome v783. Details of the reconstruction are provided in the original publications of the connectome dataset Dorkenwald et al., 2023. |
|---|---|
| Data analysis | Custom software along with widely available Python packages (e.g., numpy, scipy) were used to analyze the connectome and run simulations. Our code to reproduce main results is hosted at: https://github.com/dp4846/conn2eff |

For manuscripts utilizing custom algorithms or software that are central to the research but not yet described in published literature, software must be made available to editors and reviewers. We strongly encourage code deposition in a community repository (e.g. GitHub). See the Nature Portfolio guidelines for submitting code & software for further information.

## Data

Policy information about availability of data

All manuscripts must include a data availability statement. This statement should provide the following information, where applicable:
- Accession codes, unique identifiers, or web links for publicly available datasets
- A description of any restrictions on data availability
- For clinical datasets or third party data, please ensure that the statement adheres to our policy

*Provide your data availability statement here.*

## Research involving human participants, their data, or biological material

Policy information about studies with human participants or human data. See also policy information about sex, gender (identity/presentation), and sexual orientation and race, ethnicity and racism.

| | |
|---|---|
| Reporting on sex and gender | *Use the terms sex (biological attribute) and gender (shaped by social and cultural circumstances) carefully in order to avoid confusing both terms. Indicate if findings apply to only one sex or gender; describe whether sex and gender were considered in study design; whether sex and/or gender was determined based on self-reporting or assigned and methods used.*<br>*Provide in the source data disaggregated sex and gender data, where this information has been collected, and if consent has been obtained for sharing of individual-level data; provide overall numbers in this Reporting Summary. Please state if this information has not been collected.*<br>*Report sex- and gender-based analyses where performed, justify reasons for lack of sex- and gender-based analysis.* |
| Reporting on race, ethnicity, or other socially relevant groupings | *Please specify the socially constructed or socially relevant categorization variable(s) used in your manuscript and explain why they were used. Please note that such variables should not be used as proxies for other socially constructed/relevant variables (for example, race or ethnicity should not be used as a proxy for socioeconomic status).*<br>*Provide clear definitions of the relevant terms used, how they were provided (by the participants/respondents, the researchers, or third parties), and the method(s) used to classify people into the different categories (e.g. self-report, census or administrative data, social media data, etc.)*<br>*Please provide details about how you controlled for confounding variables in your analyses.* |
| Population characteristics | *Describe the covariate-relevant population characteristics of the human research participants (e.g. age, genotypic information, past and current diagnosis and treatment categories). If you filled out the behavioural & social sciences study design questions and have nothing to add here, write "See above."* |
| Recruitment | *Describe how participants were recruited. Outline any potential self-selection bias or other biases that may be present and how these are likely to impact results.* |
| Ethics oversight | *Identify the organization(s) that approved the study protocol.* |

Note that full information on the approval of the study protocol must also be provided in the manuscript.

# Field-specific reporting

Please select the one below that is the best fit for your research. If you are not sure, read the appropriate sections before making your selection.

☒ Life sciences  ☐ Behavioural & social sciences  ☐ Ecological, evolutionary & environmental sciences

For a reference copy of the document with all sections, see nature.com/documents/nr-reporting-summary-flat.pdf

# Life sciences study design

All studies must disclose on these points even when the disclosure is negative.

| | |
|---|---|
| Sample size | n/a |
| Data exclusions | n/a |
| Replication | Robustness studies of eigendecompisition results to noise were performed. |
| Randomization | n/a |
| Blinding | n/a |

# Reporting for specific materials, systems and methods

We require information from authors about some types of materials, experimental systems and methods used in many studies. Here, indicate whether each material, system or method listed is relevant to your study. If you are not sure if a list item applies to your research, read the appropriate section before selecting a response.

## Materials & experimental systems

| n/a | Involved in the study |
|-----|----------------------|
| ☒ ☐ | Antibodies |
| ☒ ☐ | Eukaryotic cell lines |
| ☒ ☐ | Palaeontology and archaeology |
| ☒ ☐ | Animals and other organisms |
| ☒ ☐ | Clinical data |
| ☒ ☐ | Dual use research of concern |
| ☒ ☐ | Plants |

## Methods

| n/a | Involved in the study |
|-----|----------------------|
| ☒ ☐ | ChIP-seq |
| ☒ ☐ | Flow cytometry |
| ☒ ☐ | MRI-based neuroimaging |

## Plants

| | |
|---|---|
| Seed stocks | *Report on the source of all seed stocks or other plant material used. If applicable, state the seed stock centre and catalogue number. If plant specimens were collected from the field, describe the collection location, date and sampling procedures.* |
| Novel plant genotypes | *Describe the methods by which all novel plant genotypes were produced. This includes those generated by transgenic approaches, gene editing, chemical/radiation-based mutagenesis and hybridization. For transgenic lines, describe the transformation method, the number of independent lines analyzed and the generation upon which experiments were performed. For gene-edited lines, describe the editor used, the endogenous sequence targeted for editing, the targeting guide RNA sequence (if applicable) and how the editor was applied.* |
| Authentication | *Describe any authentication procedures for each seed stock used or novel genotype generated. Describe any experiments used to assess the effect of a mutation and, where applicable, how potential secondary effects (e.g. second site T-DNA insertions, mosiacism, off-target gene editing) were examined.* |

