## [Peer Review File · Nature]

Manuscript Title: The fly connectome reveals a path to the effectome

Reviewer Comments & Author Rebuttals

Reviewer Reports on the Initial Version:

Referees' comments:

Referee #1 (Remarks to the Author):

In this manuscript, the authors present a statistical modeling framework for inferring causal relationships between neurons that minimizes confounds introduced by the impact of unobserved factors in the network. They use an instrumental variables (IV) approach to reveal the causal relationship between a “source” and “target” neuron, even in the presence of strongly correlated variability from unknown sources, by driving the source neurons with an independent source of excitation, like laser-driven optogenetic activation. They find that, when constrained by priors in the connectome, this approach can recover the “effectome” in simulations. They also provide a suggested experimental approach to measure the effectome in the fly brain.

The major strength of this work is the statistical modeling framework itself, and its promise in overcoming what is a real limitation in understanding functional relationships among neurons. To my knowledge, this approach is novel in systems neuroscience, and seems well suited for this problem. Unfortunately, some major limitations temper excitement about the work, including the lack of demonstration that the approach is useful in biological systems, and the impracticality of proposed experimental approach. After the introduction of the model, the paper reads more like a proposal than a concrete finding. Below I outline these issues and provide what I hope are useful suggestions for the authors to improve the manuscript.

Major issues and suggestions for improvement:

1. The utility of the approach in biological systems is not demonstrated.

a. The modeling framework was tested against a “ground truth” which was an imaginary effectome, defined as the connectome with some noise. Can the modeling framework recover effectomes that differ substantially from the connectome prior?

b. A major assumption used in the IV-Bayes estimator is that neurons with no synaptic contacts are unlikely to directly affect each other [line 237]. However, multisynaptic pathways and non-synaptic “wireless” communication via neuromodulatory signals can shape neural responses strongly. See, for example, a recent experimental approach to learning the “effectome” in *c. elegans* [Randi, Sharma, Dvali, Leifer, 2023]. If this assumption is relaxed, does the framework still learn the correct effectome?

c. With the effectome in hand (either complete or for a small sub-network), could this model predict brain activity given a sensory stimulus? The probe used to measure causal interactions is difficult to relate to physiological stimulation, so this remains an open question. I’m not sure how to address this question without actually learning the effectome for a small subnetwork and testing its predictive power. But it seems important before experimentalists proceed to characterize the

whole-brain effectome. A simulation of a small network may help here, as well.

2. The proposed experimental approach is very difficult at best, and likely impractical with current technologies. As one small example, extracting reliable single cell morphologies from central brain tissue requires exceptionally sparse labeling, even sparser than the predicted stochastic labeling of SPARC-S. To the extent that this paper is a “pitch” to experimentalists to test the model, I think a proposed experimental approach that focuses on a more defined circuit would be more useful, because measuring the full effectome is a monumental task, even if all of these experimental tricks work out. For example, perhaps an investigator is interested in a cell type of interest – could the authors provide a roadmap for revealing the important networks that that cell is involved in, and what the function of those circuits might be?

Referee #2 (Remarks to the Author):

A. Summary of the key results

This paper tried to understand ‘effectome’ which indicates how neurons affect each other beyond ‘connectome’ about whether neurons are connected in the fly brain. To learn a causal model of the fly brain, they first work on using optogenetics perturbation data for estimating causal effects. They indicate photostimulation could be used as “instrumental variables (IV)” that enables accurate estimation without affecting by unobserved inputs such as “confounding variables”. Later they applied connectome data as a prior distribution (IV-Bayes) to improve estimation, especially improving where the connection is zero in the prior, given the sparsity of connectome is very high (0.01%). Then they provide a roadmap to generate effectome using IV-Bayes, to provide the fraction of experiments/neurons given different choices for a number of source neurons and target neurons in photostim experiments, considering the limitation of optogenetics perturbation to image whole-brain simultaneously. They later use the eigendecomposition of connectome to identify dominant circuits in a linear dynamical system. They studied the sparsity of top-ranked eigencomponents, and visualized how neurons with top contributions in each component are localized in different fly brain areas (visual, olfactory, motor/navigation). Lastly, they show the effectome eigenvector maps related to identifiable circuits and interpretable dynamics, two examples shown in opponent motion computation and dynamic visuospatial selectivity.

B. Originality and significance: if not novel, please include reference

This paper introduces novel concepts to use connectome and linear dynamical model as a simulator to provide a roadmap for optogenetics perturbation experiments in fruit flies. They also show using connectome as a prior to improve the estimation of the causal model of the fly brain. Given the newly released full-connectome in the fruit fly, utilizing the anatomical structure information from fly connectome to guide functional experiments is well-motivated. The findings about the sparsity observed in eigendecomposition, spatial localization of eigencircuits, the discovery of identifiable circuits, and interpretable dynamics dominant by a small number of local circuits in the fly brain are novel and insightful. While their novelty is relatively limited given recent works [1][2] using connectome as prior to predict brain function, related references are also missing.

[1] Lappalainen et al. Connectome-constrained deep mechanistic networks predict neural responses across the fly visual system at single-neuron resolution, 2023.

[2] Mi et al. Connectome-constrained Latent Variable Model of Whole-Brain Neural Activity, 2022.

C. Data & methodology: validity of approach, quality of data, quality of presentation

Data: One major limitation of this work is to perform studies on simplified linear dynamical systems as simulators, instead of real functional recordings from the fly brain. It ignores the major challenges of complex network dynamics and significant trial-to-trial variability in system identification of the brain.

Method:

a. The linearity and stationary assumption contradicts biological realism. It ignores the fact of nonlinearity and synaptic plasticity observed in the brain. Given a nonlinear dynamical model, perturbing one neuron as the source each time to discover the causal matrix will no longer be an optimal solution. Extensions to perturb multiple source neurons simultaneously should be further explored. Secondly, given the existence of nonlinearity, a correlation equal to zero fails to represent the independence of two variables.

b. The connectome is only used a prior to improve the estimation of the dynamical model, while it is not used as a prior to design optogenetics stimulation strategy.

Presentation: This paper is organized in a good structure and well-written, and the results are clearly presented.

D. Appropriate use of statistics and treatment of uncertainties

a. Error bar has been properly reported in major results.

b. Natural variation across fly brains are properly modeled as corrupted versions of ground truth connectome weights.

E. Conclusions: robustness, validity, reliability

a. The results are robust and valuable given the assumption of a linear dynamical system, while their limitations are prominent when considering nonlinearity.

b. The findings about the eigencircuits are validated given some of them are also spatially localized in connectome graphs. More discussions should be introduced for some negative examples when they are not spatially localized.

c. The discovery of identifiable circuits and interpretable dynamics are aligned to some known circuit dynamics in fly brains.

F. Suggested improvements: experiments, data for possible revision

a. In Fig 2 (E), IV and IV-Bayes are not converged given the maximal samples 10^4 , more examples are needed to show when their error will converge.

b. Extension to a simple nonlinear dynamical system, $dr(t)/dt = g(Wr)$, g is a nonlinear function. Explore optogenetics stimulation patterns with more than one source neurons included each time, and explore whether connectome as a prior could improve the estimation in this scenario.

c. Show more circuits in Fig G, including negative examples where eigencircuits are not localized.

G. References: appropriate credit to previous work?

See sec 2 above.

H. Clarity and context: lucidity of abstract/summary, appropriateness of abstract, introduction, and conclusions

a. Abstract: Clarification about the optogenetic perturbation data generated from simulators that use simplified linear dynamical models instead of real recordings should be added to the abstract.

b. Introduction: Clarification about the source and confidence of the signs of all neurons and connections in connectome data.

c. Conclusions: Include more discussions of the mismatch between real brain dynamics and linear dynamical models as simulators, and discuss when the current method is still applicable and when it fails.

Referee #3 (Remarks to the Author):

The paper by Pospisil, Aragon and Pillow presents theoretical considerations and results in the fruit fly related to the estimation of synaptic connectivity based on the effects of causal manipulations — the ‘effectome’ —, and using the estimated connectome to derive functionally relevant circuits. The paper is well written —although at times the authors sacrifice some nuance for flow and simplicity; specific comments below—, and the figures are informative. Scientifically, I find their use of the Instrumental variable (IV) approach interesting, and the Bayesian extension seems to significantly improve weight estimation during the simulation scenario they consider. Adding circuit priors to their IV models and identifying dominant ‘eigencircuits’ in the fly brain are sound steps that seem to yield interesting results. I have, however, a number of concerns that should be addressed:

1. The IV approach considers independent variables that affect both the outcome (variable Y in their Fig 1B) and dependent variables (X in their Fig 1B), in this case the activity of the ‘stimulated’ neurons and the observed neurons. The authors do that in their simulations by adding variable Z in Fig 1B, but they only consider the case in which this independent variable Z is shared between the outcome measurements (Y) and dependent variables (X). This is, of course, a very reasonable first step, but there are other scenarios that should be considered in order to establish the applicability of their approach in awake behaving animals. In both cases the question would be: do the inferred weights still accurately reflect the actual weights?

1a. In the most general case, the synaptic inputs to the stimulated (X) and observed (Y) subpopulations of neurons in a behaving fly will be different, since a specific behaviour will be associated with a specific activity pattern. This ongoing activity will in turn affect the observed responses in a behaviour-specific manner (e.g., Shelchkova et al Nature Comms 2023 showed in humans that M1 responses to S1 intracortical microstimulation are task-dependent). One way to model this scenario could be to provide two different independent variables Z_x and Z_y to the stimulated and observed neurons, and systematically manipulate their relationship (e.g., the strength of their correlation/coherence), characterising the accuracy of the estimated weights.

1b. Another related scenario to simulate is the case in which the independent variables are higher dimensional activity patterns (rather than a one dimensional signal), each of which is projected to the stimulated and observed neurons via different projection matrices. These projection matrices could span a ‘relationship space’ that goes between correlated to fully independent, and ideally, with different units receiving different levels of ‘noise’. This would mimic, to some extent, the recent Leifer lab paper (Randi et al Nature 2023) showing that anatomy-derived activity predictions in the

worm perform fairly poorly if the influence of extrasynaptic communication is not considered (more on this on specific comment E below).

1c. Finally, it would be very compelling if the authors performed these simulations focusing on two functionally distinct pathways, such as the visual pathways and the motor/orientation pathways. Would there be a difference between the ability to recover the weights between functionally different structures? (I don't feel this analysis is critical, but I do think that considering more complex independent inputs as outlined in comments 1a and 1b is).

2. Figure 2: Perhaps I missed this in the Methods (which are too brief and should be significantly expanded), but the authors only seem to have validated their Bayesian IV approach on simulations in which they 'stimulated' a single neuron and recorded from the entire brain (Lines 220-21). However, later in the paper they argue that for practical and scientific reasons, stimulation should be applied to a specific population of neurons. Accordingly, the authors should validate their Bayesian IV approach when larger neural populations are simultaneously active.

3. As far as I could understand from the Methods, the authors always simulate brain dynamics as evolving according to a first order AR system. This is obviously the best first approximation, but I feel that the authors should simulate the effect of higher order dynamics on their estimations, since this would be a more realistic scenario. How would their ability to infer the actual weights be affected by considering higher order autoregressive models? (I realise that inferring the weight matrix dynamics for the second order term may be tricky, but it could be some nonlinear function of the first order term matrix)

4. The eigencircuit analysis suggests that the dynamics of the fly brain are quite high-dimensional. It could be interesting to see the effect of specific changes in circuit connectivity on this measure (as well as on the timescale analysis), since this could lead to specific predictions in certain lines. For example, what would happen if synaptic strength in specific regions is increased? Or if brain connectivity were less sparse? Or if regions were connected by more "hub" neurons?

4.1. Relatedly, I found it surprising that the first eigencircuits have very fast dynamics and involve very few neurons. The approach the authors took to arrive to this conclusion is reasonable, but what would happen if they increased the percentage of eigenvector loading power from 75 % to, e.g., 90 % or even 95 %, would their results change very significantly?

4.2. Crucially, the authors should investigate whether the inferred eigencircuits are consistent across different sets of hyper parameters (at least the loading power mentioned above)? A sensitivity analysis should be performed.

4.3. The 'not anatomically localised circuits' merit further investigation. In particular, the authors should verify how robust these circuits are to different parameter choices. Moreover, I think that their proportion could perhaps be used as a goal to be minimised —I'm not implying that the authors should develop an optimisation process to minimise this quantity, but rather that perhaps this number exhibits some relationship with specific parameters that could reveal an interesting trend.

5. The eigencircuit idea is very interesting, and deserves further investigation:

5.1. Providing a more quantitative description of their findings including the distribution of timescales, number of neurons, (spatial) distance between neurons, distribution across areas, etc.,

would be very interesting. This should be done for a reasonable number of eigencircuits, in addition to the two that are discussed in depth in Fig 5.

5.2. My understanding is that the authors performed this analysis under the assumption of linear summation of synaptic weights. This is a reasonable first order approximation but the authors should check whether nonlinear functions (e.g., an exponential) allow recovering similar eigencircuits. This feels especially important given both the account of nonlinear summation by some neuron types, and that they use linear methods for matrix factorisation to identify the eigencircuits.

6. The paper is well-written, but at times I struggled with the overarching logical flow. Could the authors confirm that my understanding is correct (and edit the manuscript for high-level clarity if they feel it's appropriate)? First, the authors show that they can estimate the connection weights of a fly using single cell stimulation and whole brain recordings provided that they add prior knowledge on fruit fly brain connectivity (Figs 1 and 2). Then they argue that stimulation of a subpopulation of neurons together with partial brain recordings is the best approach in practice (Fig 3) —a section that could be shortened, and that again calls for an extension of their simulation to this type of scenarios as I commented on #1 above. Finally, they show that eigendecomposition of the 'measured' connectome can help identify specific 'eigencircuits' and form hypotheses about their computations (Fig 4); the authors provide two such examples (Fig 5). If this reading is correct, to me, the initial emphasis on the effectome detracts from the fact that it's a means to estimating the connectome, which is the main object whose analysis can reveal interesting circuits and their computations.

7. The authors should provide more details on the Methods; I struggled finding a number of details (e.g., how were the visual inputs in Fig 5 simulated?).

SPECIFIC COMMENTS

A. Lines 63-64: the authors claim that optogenetics likely meet the criteria for being an IV but, in my view, this may only apply to very brief stimulation pulses. Long stimulation trains may put the system in 'unnatural neural states', potentially altering the 'natural' mapping between the stimuli and effects via complex changes in independent factors (e.g., saturation of certain cells, etc) that simple models won't be able to capture.

B. The authors explain in the Introduction that they are interested in the neural interactions that contribute most strongly to whole-brain dynamics (Lines 94-95). This seems very sensible —and is a more achievable aim—, but a variety of lines of evidence indicate that the largest components in neural activity may not be directly related to the production of behaviour, at least in mammals (e.g., the observation that activity in mouse V1 is mostly related to arousal/behavioural state in Stringer et al Nature 2019, the observation that most of the variance in monkey PFC during sensory-based decision making is not related to either the stimulus or the decision [Kobak ... Machens et al eLife 2016], the finding that monkey M1 is dominated by signals that do not relate to the produced movement [Churchland lab]). The question then becomes, can the method in this paper potentially

allow us to relate specific activity patterns and behaviours/sensations or is it limited to much coarser patterns? (I am not challenging the authors' core hypothesis, just pointing this out in case they want to incorporate these ideas into the paper).

C. This is not my paper and as such this is just a suggestion, but I felt that the relatively long section around Fig 3 sidetracked my thoughts when reading the paper. I don't have a specific suggestion but perhaps the authors could consider reformatting? Leaving the key idea in the discussion and appendix?

D. Figure 5D,J: The synaptic count x sign scalebar always takes very negative values, so the chosen color scale is odd. Relatedly, is this the case across all dominant modes?

E. The paper is focused on predicting neural responses from optogenetic stimulation to build a functional map of the fly brain. While it's a very interesting topic, I think that philosophically speaking a 'causal model of the brain' should be about predicting behavioural output, not just neural responses. This should be recognised in the paper.

F. Neuropeptides: Recent work in the worm from the Leifer lab (Randi et al Nature 2023) shows that non-synaptic communication (through neuropeptides) may be a large contributor to the generation of neural activity, helping to explain as much as 30 % of the total "neural variance". The authors mention their potential role as shared (between the responses of observed neurons and that of the unobserved 'outcome neurons') independent inputs, but the Leifer paper suggests that their contributions may be much higher dimensional. This should at least be discussed, and some claims in the paper softened, e.g., Lines 214-15 "Thus IV is not corrupted by unknown, unobserved inputs". The authors should also model this scenario as I indicate in comment #1 above.

Author Rebuttals to Initial Comments:

We thank the reviewers for their thorough reading of our manuscript and thoughtful feedback. Below, we summarize the primary changes we have made in response to high-level reviewer concerns. We then provide a detailed response to the concerns raised by each reviewer (in blue text), with our response and changes given below (in black and black italics, respectively). We believe the manuscript is much stronger, and hope that the reviewers will be persuaded that it is now suitable for publication.

Primary points of criticism

1. Biological realism

The reviewers had a common concern that the linear dynamical system model we used for our analyses may not be a good approximation to real neural population dynamics. We greatly appreciate this concern and are glad to have been pushed to more deeply consider the IV estimator and the connectome prior in a more realistic context.

We addressed this concern by analyzing our estimator in the context of a conductance-based model of neural dynamics. We show that the IV estimate converges to the Jacobian matrix (derivative w.r.t. voltage) of the underlying neural dynamics equation, evaluated at the average voltages of the neurons in the population. The i,j th entry in the Jacobian predicts the effect of a unit perturbation of the j th neuron on the i th neuron—the accuracy of this prediction can achieve arbitrary precision as the magnitude of the perturbation decreases. This result is entirely consistent with our notion of the effectome: a matrix that predicts the effect of every neuron on every other neuron. Our linear model is a special case where the Jacobian is proportional to the connectome (synaptic count and sign). In a nonlinear model, the effect of a perturbation depends on the state of the network; the Jacobian (which also depends on the state of the network) captures this notion exactly. The Jacobian is a foundational quantity in the study of stochastic differential equations and is classically employed to obtain a linear approximation to a nonlinear dynamical system in a particular operating regime. We feel this result improves the interpretability of our estimator when applied to nonlinear neural dynamics and furthermore its utility given the Jacobian is a well developed tool in the study of stochastic dynamics (e.g., for system approximation and control). To emphasize this result we now define the effectome as the Jacobian of underlying neural dynamics and motivate its estimation in our new Discussion section: *'Interpretation of IV based LDS estimator applied to nonlinear dynamics'*.

The reviewers also expressed concern that the connectome prior may be systematically wrong in a realistic biological context. Our new interpretation of our estimator as providing the Jacobian of neural dynamics further clarifies the utility of the connectome prior. We show analytically that in a conductance-based model of neural dynamics, the Jacobian (local linear effect) between unconnected neurons will always be zero. Thus the connectome prior mean will only be wrong for 0.04% of the effectome parameters because only 0.04% of pairs of neurons form a synapse or more in the fly brain. Thus, a loss of efficiency from misspecified non-zero weights is minimal. We also show that even with a misspecified prior (e.g., effectome weights synapses may not be proportional to the number of synapses, or there may be extra-synaptic effects) our estimator, with enough data, converges to the true Jacobian. In general, most neurons in the fly brain do not directly affect each other and this is where the majority of our

gains in statistical efficiency are achieved. This in combination with weak priors on the small subset of neurons that plausibly directly affect each other allow our estimator the flexibility to efficiently estimate the effectome. We illustrate this consistency in both nonlinear and linear dynamics with a new set of simulations, which we provide in two new sections of the supplement, 'IV estimator applied to nonlinear dynamics' and 'IV estimator additional simulations' (SI Fig 3-6).

2. More thorough characterization of eigencircuits and their robustness.

The reviewers agreed that the concept of eigencircuits is interesting, and that our findings are intriguing, but felt that analyses of eigencircuits could be more thorough in two ways.

(A) Including summary analyses of the joint relationships between anatomy, eigencircuits, and time scales, with a focus on the non-localized eigencircuits.

We have now thoroughly examined marginal and joint relationships of these properties. Some new findings of interest are:

- Anatomically localized eigencircuits (forming most synapses in a single neuropil) represent a minority of the eigencircuits we discovered (making up <10 % of top 1000 eigencircuits); the majority of localized eigencircuits are in the top ~200 eigencircuits.
- There are three distinct peaks in the distribution of timescales: most eigenvalues have a complex component that gives rise to oscillations with a period of 4; second-most common are eigenvalues with a period of 2 (the fastest possible oscillation) which tended to be circuits with strong reciprocal inhibition; finally, a smaller number of eigenvalues were real and positive corresponding to slow monotonic decay, which tended to be circuits with strong reciprocal excitation.
- Dominant eigencircuits tended to have neuron pairs with larger numbers of synapses but overall fewer synapses in the entire eigencircuit (because of their higher sparsity).

(B) Characterizing the robustness of the discovered eigencircuits.

We have now characterized the robustness of discovered eigencircuits to measurement error by adding noise to the original connectome synapse counts and determining if the same eigencircuits are recovered. We find that even when half of the variance across synaptic counts is noise the eigencircuits remain robust (SI Fig 9). We found a tendency for lower and less local eigencircuits to be corrupted by noise more easily (SI Fig 10).

We also characterize their robustness by re-running our analyses under different nonlinear transformations between connectome and effectome. In particular, we calculated eigencircuits of a connectome where we applied a saturating function (hyperbolic tangent) to each entry. Thus the magnitude of effects between neurons saturates as a function of the number of synapses. In general we found a gradual degradation of the original eigencircuits as we increased the effect of the nonlinearity (SI Fig 10). Similar to the case of adding noise to the connectome, lower eigencircuits tended to be corrupted more easily.

We have now included all new results described above in new sections of the supplement and reference them in the main text where relevant.

Referee #1 (Remarks to the Author):

In this manuscript, the authors present a statistical modeling framework for inferring causal relationships between neurons that minimizes confounds introduced by the impact of unobserved factors in the network. They use an instrumental variables (IV) approach to reveal the causal relationship between a “source” and “target” neuron, even in the presence of strongly correlated variability from unknown sources, by driving the source neurons with an independent source of excitation, like laser-driven optogenetic activation. They find that, when constrained by priors in the connectome, this approach can recover the “effectome” in simulations. They also provide a suggested experimental approach to measure the effectome in the fly brain.

The major strength of this work is the statistical modeling framework itself, and its promise in overcoming what is a real limitation in understanding functional relationships among neurons. To my knowledge, this approach is novel in systems neuroscience, and seems well suited for this problem. Unfortunately, some major limitations temper excitement about the work, including the lack of demonstration that the approach is useful in biological systems, and the impracticality of proposed experimental approach. After the introduction of the model, the paper reads more like a proposal than a concrete finding. Below I outline these issues and provide what I hope are useful suggestions for the authors to improve the manuscript.

Major issues and suggestions for improvement:

1. The utility of the approach in biological systems is not demonstrated.

a. The modeling framework was tested against a “ground truth” which was an imaginary effectome, defined as the connectome with some noise. Can the modeling framework recover effectomes that differ substantially from the connectome prior?

We thank the reviewer for this comment, and agree that it is important to demonstrate that our method will work in biologically realistic settings (including where the effectome differs substantially from the mean of the connectome prior). To address the reviewer’s comment, we would first like to point out that: yes, the estimator will indeed converge to the true effectome even if the connectome prior is mismatched to the true effectome. This is guaranteed by the fact that (for Bayesian methods in standard settings) the data ultimately “overwhelms the prior”, and the method converges to ground truth parameters given enough data. Yet, a small-variance prior (i.e., a highly confident prior) will take a larger amount of data to overwhelm the prior. To demonstrate this, we ran a small simulation where the ground truth effectome was independent of the connectome prior. We generated a sparse ground truth effectome that we also used as the ‘correct’ connectome prior mean, and to assess the effect of a misspecified prior we independently sampled a sparse ‘incorrect’ connectome prior mean (Response Fig 1A). We then applied IV-Bayes with the correct and incorrect connectome prior means (B) with an increasing constant added to the prior variance (left to right).

IV-Bayes is a consistent estimator even with misspecified prior

SI Fig. 5: IV-Bayes is a consistent estimator even with incorrect prior **(A) (left)** Example ground truth simulated effectome for linear simulations (off-diagonal weights drawn from uniform distribution $([0.1,0.2])$, then on average 9- % are set to 0 and diagonal is set to 0.1) **(A) (center)** In ‘Correct prior’ condition prior mean is set to the true effectome. **(A) (right)** In ‘Incorrect prior’ condition prior mean is set to independent effectome (without diagonal). **(B)** Simulations of IV-Bayes estimator with correct (blue) and incorrect prior (orange) and the raw IV estimator (green) across number of samples (i.e., duration of recording) and resamplings of effectomes. Variance of prior mean has a constant (see Fig 2C legend) of increasing size added to it (increasing left to right). R^2 is measured between the estimate of the effectome and ground truth.

We find that when the connectome prior variance is small (B left) the estimator with the correct prior (blue) converges more rapidly than the naive estimate with no prior (green) and far faster than the estimator with the incorrect prior (orange). As the prior variance increases (B left to right) the estimator with the incorrect prior becomes more efficient while the estimator with the correct prior becomes less efficient. We now emphasize this in the introduction to the results (line # 154):

‘This motivates using the connectome as a prior on the IV weights so that the estimator remains consistent (*i.e., even if the prior is wrong the estimator will converge to ground truth with enough data*) yet potentially orders of magnitude more efficient—to the degree the prior is correct.’

b. A major assumption used in the IV-Bayes estimator is that neurons with no synaptic contacts are unlikely to directly affect each other [line 237]. However, multisynaptic pathways and non-synaptic “wireless” communication via neuromodulatory signals can shape neural responses strongly. See, for example, a recent experimental approach to learning the “effectome” in *C. elegans* [Randi, Sharma, Dvali, Leifer, 2023]. If this assumption is relaxed, does the framework still learn the correct effectome?

The reviewer is correct and raises an important point: there are many extra-synaptic paths by which neurons can affect each other and the connectome prior, being purely synapse based, does not explicitly account for them. However, the IV-Bayes framework gracefully incorporates uncertainty about whether the absence of a synapse implies there is no direct effect between neurons. While the mean of the connectome prior is zero for two neurons without a synapse, the prior variance quantifies uncertainty about whether there really is no effect. Because our prior variance is non-zero, if there is a linear effect between neurons despite no measured synapse (either because it was missed in the connectome or the effect is not through a synapse) the magnitude and sign of that effect will be consistently estimated (as shown in SI Fig 5 above), it will simply take more samples to converge (roughly in inverse proportion to the connectome prior variance). Prior variance can be set either from an informed guess (e.g., extrasynaptic peptidergic effects are more likely between certain neuronal subtypes) or estimated from the data via cross-validation. We now recapitulate and address the reviewer’s important point in our manuscript along with simulations that demonstrate the estimator will converge to the correct effectome (line # 812):

We also confirm that even with a completely misspecified prior (e.g., there may be extra-synaptic effects through peptide signaling pathways \cite{randi_neural_2023}) our estimator, with enough data, converges to the Jacobian, that is, the weight matrix providing a linear description of the effect of each neuron on any other in the system’s current state (SI Fig 5).

The reviewer also raises the point that important effects between neurons can be multi-synaptic. This highlights the critical caveat that our estimator requires an explicit time scale for estimated effects. The effectome estimator as presented here specifically only estimates mono-synaptic effects by assuming a fixed conduction delay between neurons, in our simulations 1 ms. If there are effects slower than this time scale they will be missed (e.g., a polysynaptic effect through unobserved neurons). In our discussion we emphasize the importance of extension of our estimator to the AR(P) case to be able to model effects at different delays (line # 718):

An extension to AR(P) would relax the restriction that the time scale of interaction is known and fixed (in our simulations 1 ms) thus slower effects (potentially through extra-synaptic paths) could be detected.

We also provide supplementary simulations demonstrating this effect in higher order AR(P) models (see SI Fig 6)

This point also motivated us to emphasize that estimates of multi-synaptic effects can actually be derived from the estimated monosynaptic effects. We now raise this point in methods (line # 982)

We note that multi-synaptic effects can be derived from the estimated monosynaptic effects with powers of the effectome matrix. For example if we have the effectome matrix W and input r is an input vector of all zeros except for neuron i then the n th order synaptic effect is exactly $W^n r$ (e.g., $n=1$ gives direct synaptic effects, $n=2$ effects through up to two synapses, etc).

c. With the effectome in hand (either complete or for a small sub-network), could this model predict brain activity given a sensory stimulus? The probe used to measure causal interactions is difficult to relate to physiological stimulation, so this remains an open question. I'm not sure how to address this question without actually learning the effectome for a small subnetwork and testing its predictive power. But it seems important before experimentalists proceed to characterize the whole-brain effectome. A simulation of a small network may help here, as well.

The reviewer raises a critical question: how can our estimated model of neural dynamics be used to predict effects of external inputs? Without this ability its utility clearly comes into question. Ultimately for our dynamics model to form such a prediction requires that the effect of the stimuli on the nervous system be specified.

For example, in the case of the visual system the responses of ommatidia to arbitrary stimuli are well characterized, thus by estimating the ommatidia effects on their downstream synaptic contacts we can form predictions of arbitrary stimuli on brain dynamics. This does not guarantee that the predictions will be accurate—it is only to the degree that our parameters have been accurately estimated and our linear model is a good approximation to the conditions of stimulation.

In our simulations we used prior experimental knowledge about the relationship between stimuli and the responses of lobula plate neurons to make explicit predictions about the mechanisms and effects of stimulation, which generated results consistent with physiological recordings (Fig 5K). We hope this addresses the reviewer's concern by showing that — at least in cases where sensory response properties are known — the model can indeed be used to predict the neural dynamics resulting from sensory stimuli.

We now raise this critical point in the discussion and emphasize the critical nature of characterizing the sensory periphery (line # 982).

We were able to recapitulate experiments on the visual system by directly stimulating neurons in the central nervous system (Fig 5) but this required prior knowledge of how specific sets of stimuli affect the central nervous system. Lappalainen et al (2023) ran simulations directly from raw visual inputs via a model of light's effects on ommatidia. Incorporating the effects of stimuli on the peripheral nervous system is a critical direction for utilizing effectome estimates to predict sensory computations for novel stimuli.

2. The proposed experimental approach is very difficult at best, and likely impractical with current technologies. As one small example, extracting reliable single cell morphologies from central brain tissue requires exceptionally sparse labeling, even sparser than the predicted stochastic labeling of SPARC-S. To the extent that this paper is a “pitch” to experimentalists to test the model, I think a proposed experimental approach that focuses on a more defined circuit would be more useful, because measuring the full effectome is a monumental task, even if all of these experimental tricks work out. For example, perhaps an investigator is interested in a cell type of interest – could the authors provide a roadmap for revealing the important networks that that cell is involved in, and what the function of those circuits might be?

We thank the reviewer for requesting a concrete actionable approach to implementing the estimator proposed in this paper. We provide an example where we believe this approach is particularly feasible wherein neurons are spatially segregated and overlap with an existing GAL4 line (VT014336) with high overlap with eigenvector 25. We describe the roadmap to identifying this GAL4 line and the following experimental approach required to estimate the effectome in the new SI section 'Experimental approach to learning the effectome' (line # 1292)

Here we provide a concrete example of selecting GAL4 driver lines based on the putative effectome. Eigenvector 25 contains neurons within the antennal lobe, mushroom body, and lateral horn (Fig S12); the spatial segregation of these regions makes this circuit amenable to our proposed experimental strategy. To begin, we determine the neurons in this circuit with the highest loadings: these neurons include DP1m-adPN, v2LN30, VL2p-adPN, and DP1l-adPN. Next, we find candidate GAL4 lines that contain hits for these neurons of interest using publicly available resources \cite{meissner_searchable_2023}. We find that the Gal4 line VT014336 contains strong pixel overlap scores for each of these neurons, and is thus a suitable candidate for uncovering the causal interactions of neurons within eigenvector 25.

In addition the new 'Experimental approach to learning the effectome' section (2,500 words) streamlines and more thoroughly fleshes out our proposed strategies for addressing the challenges the reviewer raises. To more generally address the reviewers accurate assessments of the challenges of estimating the effectome we now end this section as follows (line # 1386):

It is difficult to predict how this particular experimental approach will perform in any given circuit or experimental condition without actually attempting it. This approach can at minimum serve as a starting point that uses existing technologies and upon it, if needed, iterative technological enhancements can be developed to estimate the effectome.

Referee #2 (Remarks to the Author):

A. Summary of the key results

This paper tried to understand `effectome' which indicates how neurons affect each other beyond `connectome' about whether neurons are connected in the fly brain. To learn a causal model of the fly brain, they first work on using optogenetics perturbation data for estimating

causal effects. They indicate photostimulation could be used as “instrumental variables (IV)” that enables accurate estimation without affecting by unobserved inputs such as “confounding variables”. Later they applied connectome data as a prior distribution (IV-Bayes) to improve estimation, especially improving where the connection is zero in the prior, given the sparsity of connectome is very high (0.01%). Then they provide a roadmap to generate effectome using IV-Bayes, to provide the fraction of experiments/neurons given different choices for a number of source neurons and target neurons in photostim experiments, considering the limitation of optogenetics perturbation to image whole-brain simultaneously. They later use the eigendecomposition of connectome to identify dominant circuits in a linear dynamical system. They studied the sparsity of top-ranked eigenvectors, and visualized how neurons with top contributions in each component are localized in different fly brain areas (visual, olfactory, motor/navigation). Lastly, they show the effectome eigenvector maps related to identifiable circuits and interpretable dynamics, two examples shown in opponent motion computation and dynamic visuospatial selectivity.

B. Originality and significance: if not novel, please include reference

This paper introduces novel concepts to use connectome and linear dynamical model as a simulator to provide a roadmap for optogenetics perturbation experiments in fruit flies. They also show using connectome as a prior to improve the estimation of the causal model of the fly brain. Given the newly released full-connectome in the fruit fly, utilizing the anatomical structure information from fly connectome to guide functional experiments is well-motivated. The findings about the sparsity observed in eigendecomposition, spatial localization of eigencircuits, the discovery of identifiable circuits, and interpretable dynamics dominant by a small number of local circuits in the fly brain are novel and insightful. While their novelty is relatively limited given recent works [1][2] using connectome as prior to predict brain function, related references are also missing.

[1] Lappalainen et al. Connectome-constrained deep mechanistic networks predict neural responses across the fly visual system at single-neuron resolution, 2023.

[2] Mi et al. Connectome-constrained Latent Variable Model of Whole-Brain Neural Activity, 2022.

We thank the reviewer for pointing out these highly relevant citations, and we apologize for overlooking them in our original manuscript. We have now included them in our new discussion section ‘*Related connectomic work*’ where we highlight differences but also specify the many innovative contributions in these pioneering works that could be incorporated as extensions into the IV estimator approach.

In an alternative connectome based approach applied to the visual system of the fly \cite{lappalainen_connectome-constrained_2023} parameters were not constrained to neural activity but instead optimized to perform discrimination of visual motion. By ‘sharing’ parameters across circuits thought to perform similar functions the efficiency of estimation was greatly increased. In a similar approach, IV-Bayes could be extended to a hierarchical Bayesian model that borrows statistical power across circuits and cell-types hypothesized to have similar functional properties.

We were able to recapitulate experiments on the visual system by directly stimulating neurons in the central nervous system (Fig 4) but this required prior knowledge of how specific sets of stimuli affect the central nervous system. Lappalainen et al (2023) ran simulations directly from raw visual inputs via a model of light's effects on ommatidia. Incorporating the effects of stimuli on the peripheral nervous system is a critical direction for utilizing effectome estimates to predict sensory computations for novel stimuli.

We note that a novelty of our work relative to these estimators (as recognized by the reviewer in their introduction) is that our instrumental-variable based framework has the important property that it is robust to confounding variables, such as shared inputs from other brain regions (which are likely to be ubiquitous in the brain). A subtler difference is that our connectome constraints are not 'hard' (i.e., in Lappalainen et al. neurons without anatomic connectivity cannot directly affect each other in the model) but instead probabilistic where even if anatomy is an incorrect account of the effects between neurons our estimator will converge to the correct model with enough data.

C. Data & methodology: validity of approach, quality of data, quality of presentation

Data: One major limitation of this work is to perform studies on simplified linear dynamical systems as simulators, instead of real functional recordings from the fly brain. It ignores the major challenges of complex network dynamics and significant trial-to-trial variability in system identification of the brain.

Method:

a. The linearity and stationary assumption contradicts biological realism. It ignores the fact of nonlinearity and synaptic plasticity observed in the brain. Given a nonlinear dynamical model, perturbing one neuron as the source each time to discover the causal matrix will no longer be an optimal solution. Extensions to perturb multiple source neurons simultaneously should be further explored. Secondly, given the existence of nonlinearity, a correlation equal to zero fails to represent the independence of two variables.

We greatly appreciate this concern and are grateful to have been pushed to more deeply consider the IV estimator in a more realistic context. We found that the IV estimator converges to an optimal linear approximation to a nonlinear dynamical system: the Jacobian of neural dynamics. The Jacobian is a foundational quantity in the study of stochastic differential equations and is classically employed in a linear dynamical system to approximate nonlinear dynamics along an estimated trajectory (e.g., stimulus triggered average). We feel this result improves the interpretability of our estimator when applied to nonlinear neural dynamics and furthermore its utility given the Jacobian is a well developed tool in the study of stochastic dynamics (e.g., for system approximation and control).

This finding has direct bearing on the two important concerns the reviewer raises: nonlinearity and nonstationarity (e.g., plasticity can change effects between neurons over time).

Nonlinearity: We have now provided analyses of IV in the context of a nonlinear conductance based neural dynamics model where multiple neurons are stimulated as the reviewer requested

(SI Fig 3,4). We provide a precise interpretation of IV's estimand: the Jacobian of underlying nonlinear neural dynamics typically used a linear approximation to those dynamics (for an overview see the introduction of this response above). On the one hand a linear approximation cannot capture the nonlinear dynamical repertoire of the fly nervous system. Yet, the form of neural dynamics in the fly is unknown and even for our simple conductance model the parameters needed to constrain it are difficult to measure. The Jacobian on the other hand is well defined for a broad class of neural dynamics (analytic functions), we show we can accurately estimate it, and linear dynamics have been used to great success in interpreting nonlinear dynamics (Strogatz, 1994; Newman, 2010). Furthermore it has been shown neurons often do operate in linear regimes (Ahmed et al., 1998; Chance et al., 2002). Despite the lack of biological realism of the linear approximation we believe that the huge amount of theory that has been developed to use linear approximations to understand nonlinear dynamics provides very strong support for the use of our estimator—especially given the new connection we have made between it and the Jacobian the theoretically optimal linear approximation. We thank the reviewer for motivating us to make such an important connection.

Nonstationarity: The reviewer raises the important point that a linear model cannot capture the non-stationary dynamics of the fly. We now emphasize this point in our discussion and reference our nonlinear neural dynamics simulations where we show that depending on the average voltage the effects between neurons can change (compare Supplemental Fig 3 C and D). To make this phenomena concrete we precisely quantify it with the Jacobian of the conductance model (SI eq 3) and discuss how variation in voltage over time can lead to changes in effects between neurons over time. We now emphasize this caveat in our results and how it provides excellent motivation for extensions of our method to the setting of switching linear dynamical systems which can capture non-stationary (and nonlinear) dynamics. Furthermore, with the relationship of the effectome to the Jacobian clear we also raise the point that the technique of linearizing around a *trajectory* via a time varying Jacobian can be borrowed from linear systems theory. We provide a concrete example in sensory experiments where independent IV estimates can be formed across a stimulus triggered average providing a direct window into non-stationary dynamics employed during sensory computations. We emphasize that more generally extensions to our estimator can condition IV estimates on experimental conditions, behavior, or inferred latent states.

We discuss these points in our new Discussion section '*Interpretation of IV based LDS estimator applied to nonlinear dynamics*' and in our new SI section '*IV estimator applied to nonlinear dynamics: Continuous conductance based neural dynamics model*'.

Through these additional simulations and discussion of how to interpret our estimator in a biophysically motivated nonlinear and nonstationary setting we hope the readers will have a more concrete sense of both the utility and the limitations of our approach—which can jointly motivate further research into the extensions of our method that we outline (line # 790):

The Jacobian is also a foundational quantity in the study of stochastic differential equations. It is classically employed in a linear dynamical system that approximates

nonlinear and non-stationary dynamics along an estimated trajectory. For example independent IV estimates can be formed across a stimulus triggered average. More generally, extensions to our approach can be applied to estimated trajectories within repeated experimental conditions, stereotyped behaviors, or inferred latent states (e.g., Linderman et al., 2017). Ultimately we interpret effectome estimates as capturing local interactions between neurons in a particular state (e.g., at a set of membrane voltages). Further research is needed to understand how to use these local estimates to learn about changes in state (e.g., synaptic plasticity).

b. The connectome is only used a prior to improve the estimation of the dynamical model, while it is not used as a prior to design optogenetics stimulation strategy.

The use of the connectome as a prior to design an optogenetic stimulation strategy is an excellent idea and we now include this potential extension in our discussion (line # 705).

A formal counter-factual based framing of optogenetic effects on behavior was recently provided by \cite{levis_causal_2024-1} along with methods to account for closed-loop optogenetic stimulation. Extending this technique to the setting of neural dynamics could provide a principled approach to closed-loop estimation of mechanistic neural dynamics models that leverages the connectome prior and its updated posterior (upon the observation of experimental data)

We now clarify that we are proposing to use the connectome to explicitly recommend candidate neurons for optogenetic stimulation. Specifically, we recommend first stimulating the neurons with the highest magnitude eigenvector loadings in the eigenvectors associated with the highest magnitude eigenvalues (line # 374):

We update the beginning of results section 'Identifying dominant circuits using the connectome':
Here we demonstrate a data-driven method for ranking sets of source neurons most likely to form circuits with a large effect on the fly nervous system.

To:

*Here we demonstrate a data-driven method for ranking sets of source neurons most likely to form circuits with a large effect on the fly nervous system. **We propose that these circuits should be prioritized for interrogation by our estimator.***

Presentation: This paper is organized in a good structure and well-written, and the results are clearly presented.

D. Appropriate use of statistics and treatment of uncertainties

a. Error bar has been properly reported in major results.

b. Natural variation across fly brains are properly modeled as corrupted versions of ground truth connectome weights.

E. Conclusions: robustness, validity, reliability

- a. The results are robust and valuable given the assumption of a linear dynamical system, while their limitations are prominent when considering nonlinearity.
- b. The findings about the eigencircuits are validated given some of them are also spatially localized in connectome graphs. More discussions should be introduced for some negative examples when they are not spatially localized.

We were also intrigued by these non-localized eigencircuits. We identified three examples that are diverse in terms of neuropil distribution and dynamics. We now provide examples of these circuits in SI Fig 16 (for higher resolution see manuscript):

SI Fig. 16: Examples of non-localized eigencircuits. **(A)** Circuit with neurons in the optic lobes and central complex. **(B)** Circuit with neurons in the optic lobe, antennal lobes, lateral horn, and lateral accessory lobe. **(C)** Circuit with neurons in the optic lobe and pre-motor regions.

We also discuss their properties in the supplement section titled 'Additional connectome analyses' (line # 2094)

Our eigencircuit analysis also revealed the presence of non-localized circuits that became more prevalent with increasing eigenvalue rank. A sample of these non-localized circuits reveals diverse innervation patterns \label{fig_S16}. Eigenvector 59 contains neurons that innervate both optic lobes and the central complex (Fig S13A). This circuit has a real negative eigenvalue, indicating inhibition under rectified dynamics, which may permit selective integration of visual input from a single eye depending on the strength of visual input. We also found circuits that involve multiple sensory modalities, such as eigenvector 67 (Fig S16B), which contains visual neurons in the optic lobe and olfactory neurons in the antennal lobe and lateral horn. This circuit has a complex eigenvalue, which suggests oscillatory dynamics. Interestingly, the loadings for this eigenvector have different signs (data not shown), indicating that these neurons oscillate out of sync. This asynchronous oscillation may play a role in integrating sensory information across modalities. Finally, we consider eigenvector 76, with neurons primarily located in visual and pre-motor areas (Fig S16C). This circuit is also associated with a complex eigenvalue, suggesting oscillatory dynamics. Given this circuit's

incorporation of both visual neurons and descending output neurons, we speculate that this circuit may be involved in coordinating visually-driven motor programs.

c. The discovery of identifiable circuits and interpretable dynamics are aligned to some known circuit dynamics in fly brains.

F. Suggested improvements: experiments, data for possible revision

a. In Fig 2 (E), IV and IV-Bayes are not converged given the maximal samples 10^4 , more examples are needed to show when their error will converge.

We thank the reviewer for this suggestion. Because we plotted MSE on a log axis the error will never appear to converge, the estimates can be made arbitrarily precise because they are both consistent estimators. We now clarify this in the legend of Fig 2 E:

We update:

... but IV-Bayes gives several orders of magnitude faster convergence.

To:

*... but IV-Bayes gives several orders of magnitude faster convergence. **MSE will decrease indefinitely for both estimators because they are consistent (i.e., converge to ground truth as the number of samples goes to infinity)***

b. Extension to a simple nonlinear dynamical system, $dr(t)/dt = g(Wr)$, g is a nonlinear function. Explore optogenetics stimulation patterns with more than one source neurons included each time, and explore whether connectome as a prior could improve the estimation in this scenario.

We thank the reviewer for this excellent suggestion that led to deeper insights into the IV method that greatly improved the manuscript. We have now run conductance based simulations with IV applied to all recorded neurons in a small population (new supplementary section *IV estimator applied to nonlinear dynamics*). We find that IV converges to the Jacobian of the dynamics equation evaluated at average neuronal voltage (SI Fig 3 and eq 3). The Jacobian is often used as the weights of a linear dynamical system approximating a nonlinear one. Thus given this useful estimand we then asked whether the connectome prior (synapse count and sign appropriately scaled) can improve its estimation. We found that in the context of sparse connectivity of the conductance model the connectome prior greatly improved estimation efficiency (SI Fig 4). This is despite the connectome prior and ground truth effectome not being proportional to each other—variation in average voltage across neurons leads to variation in the degree of effects between neurons (for summary see introduction of response to reviewers). This misspecification of the prior is ameliorated by neurons with no synapses between them, in this case the connectome prior and Jacobian prior align; intuitively, this is because without a synapse there is no local linear effect. Thus similarly to our whole brain simulations the connectome prior on unconnected neurons is what primarily drives gains in efficiency. We feel

that demonstrating this benefit of our estimator is retained in a more biologically realistic context strengthens the appeal of our approach.

IV converges to Jacobian of membrane voltage equation

SI Fig. 3: Instrumental variables estimator converges to the Jacobian of membrane voltage equation. **(A)** Conductance matrix, W_0 , of conductance simulation where off-diagonals were set randomly according to a uniform distribution between 0.01 and 0.02. The reversal potential of all synapses is 0 mV, thus all synapses are typically excitatory. Membrane constant $\tau = 10$ ms, membrane resistance $R = 1$, input noise is $SD=1$ mV, laser perturbation $SD=1$ mV, inputs were hand chosen to induce two conditions of similar vs different average voltages, and the conductance equation is integrated using Euler's method with $\Delta = 1$ ms with noise and laser perturbation added at each time step. **(B)** The firing rate as a function of membrane voltage is sigmoidal. **(C) (top)** Example traces of firing rate over time for neurons with similar average voltage. Different colors represent the five different neurons. **(C) (bottom left)** Relationship between IV estimates and the Jacobian evaluated at the average voltage, with a high correlation ($r = 0.96$) and a linear fit with slope near 1 indicating IV converges to the Jacobian of neural dynamics. **(C) (bottom right)** Relationship between the Jacobian and conductances (W_0), with a strong correlation ($r = 0.96$) and slope deviating from one indicating that approximately $J \propto W$. **(D) (top)** Example traces of firing rate over time for neurons with different average voltages. **(D) (bottom left)** IV estimates converges to Jacobian again. **(D) (bottom right)** In contrast Jacobian and conductance matrix are not proportional to each other when average voltages varies widely.

IV-Bayes improves efficiency of Jacobian estimation in conductance model

SI Fig. 4: IV-Bayes improves the efficiency of Jacobian estimation in the conductance model. **(A)(left)** Example of sparse synaptic connectivity matrix W_0 used for conductance simulations, where off-diagonals were set randomly according to a uniform distribution between 0.01 and 0.02, then on average 90 % of these were set to 0. **(A)(right)** Example firing rate traces across 10 neurons. **(B)** Comparison of IV and IV-Bayes estimate of Jacobian. **(C)** Comparison of R^2 values for IV and IV-Bayes estimators as a function of the number of samples. The plot shows that the IV-Bayes estimator achieves higher R^2 values with fewer samples compared to the IV estimator. Error bars show standard deviations across 5 simulations (a different W_0 chosen for each simulation but all other parameters remain the same).

c. Show more circuits in Fig G, including negative examples where eigencircuits are not localized.

Due to space constraints, we now include additional eigencircuits in the supplementary figures. SI Fig 12-15 shows the first 100 unique eigencircuits, that include many non-localized circuits. We include cell labels, eigenvalues, and indices into Flywire to facilitate their further

investigation (for higher resolution see manuscript):

SI Fig. 12: First 25 eigencircuits based on the putative effectome (eigencircuits 1 to 25). Circuits are ordered by increasing eigenvalue rank.

SI Fig. 14: Third set of 25 eigencircuits based on the putative effectome (eigencircuits 51 to 75).

SI Fig. 13: Second set of 25 eigencircuits based on the putative effectome (eigencircuits 26 to 50).

SI Fig. 15: Fourth set of 25 eigencircuits based on the putative effectome (eigencircuits 76 to 100).

SI Fig 16 shows three representative non-localized circuits. which are discussed in the supplemental section 'Additional connectome analyses.'

G. References: appropriate credit to previous work?See sec 2 above.

We now cite these references in the discussion (line # 747)

Anatomical information has been used to constrain mechanistic whole brain models of the worm fit to neural data \cite{mi_connectome-constrained_2021} and a similar model could be applied to the fly. A critical distinction of our approach is that it provides consistent estimates under arbitrary confounding noise. Even for whole brain recordings, now possible in the worm, where one might assume there are no unobserved variables, it is impossible to directly test this assumption. Yet, this pioneering work directly demonstrates that connectomic constraints can improve the efficiency of model estimation--as we show in simulation. In an alternative connectome based approach applied to the visual system of the fly \cite{lappalainen_connectome-constrained_2023} parameters were not constrained to neural activity but instead optimized to perform discrimination of visual motion. By `sharing' parameters across circuits thought to perform similar functions the efficiency of estimation was greatly increased. In a similar approach, IV-Bayes could be extended to a hierarchical Bayesian model that borrows

statistical power across circuits and cell-types hypothesized to have similar functional properties.

H. Clarity and context: lucidity of abstract/summary, appropriateness of abstract, introduction, and conclusions

a. Abstract: Clarification about the optogenetic perturbation data generated from simulators that use simplified linear dynamical models instead of real recordings should be added to the abstract.

Thanks for this suggestion. We have now updated the abstract from:

Specifically, we propose an estimator for a dynamical systems model of the fly brain that uses stochastic optogenetic perturbation data to accurately estimate causal effects and the connectome as a prior to drastically improve estimation efficiency.

To:

*Specifically, we propose an estimator for a **linear** dynamical systems model of the fly brain that uses stochastic optogenetic perturbation data to accurately estimate causal effects and the connectome as a prior to drastically improve estimation efficiency. **We validate our approach in whole brain connectome-based simulations.***

b. Introduction: Clarification about the source and confidence of the signs of all neurons and connections in connectome data.

We have now updated the introduction (line # 115) from:

Furthermore, with synapse-level neurotransmitter predictions as provided by the Flywire connectome \cite{eckstein_neurotransmitter_2023}, priors can be placed on the sign of effects between the neurons that are connected (the confidence of those predictions can naturally be incorporated into the strength of the prior).

To:

*Furthermore, with synapse-level **EM based** neurotransmitter predictions as provided by the Flywire connectome \cite{eckstein_neurotransmitter_2023} (**see Methods, 'Construction of fly connectome matrix'**), priors can be placed on the sign of effects between the neurons that are connected (the confidence of those predictions can naturally be incorporated into the strength of the prior).*

We also included a new section in the methods which we reference in the introduction (line #1029):

Construction of fly connectome matrix

The connectome is a reconstruction of the central nervous system of a 7 day old adult female \textit{Drosophila melanogaster}. We use the most recent version of the connectome v783. Details of the reconstruction are provided in the original publications of the connectome dataset \cite{dorkenwald_neuronal_2023-1}.

Each entry in the connectome matrix W , the main object of study in our analyses, was the number of synapses multiplied by their inferred sign based on predicted neurotransmitter type \cite{eckstein_neurotransmitter_2023}. Specifically, neurons with neurotransmitters acetylcholine and dopamine had positive weights on their downstream

neurons and neurons with GABA, serotonin, glutamate, and octopamine had negative weights. neurotransmitter was predicted directly from EM images trained on synapses with known neurotransmitter types. The matrix W scaled for stability was used as the connectome prior mean in estimator simulations (Fig \ref{fig2}) and our eigendecomposition analysis (eq. \ref{2}; Fig. \ref{fig3}, \ref{fig4}). A threshold was set on the synapse count such that any connections with less than 5 synapses were set to 0. This choice followed the reasoning of other analyses of the connectome \cite{lin_network_2023} that this would minimize the impact of spurious synapses---manual proofreading did not extend to connections with fewer than 5 synapses.

c. Conclusions: Include more discussions of the mismatch between real brain dynamics and linear dynamical models as simulators, and discuss when the current method is still applicable and when it fails.

We thank the reviewer for encouraging us to more deeply consider our estimator in the context of biologically realistic neural dynamics. We have now conducted analyses of our estimator in the context of a conductance model of neural dynamics that provide a concrete interpretation of our estimator applied to nonlinear dynamics (new SI section ‘*IV estimator applied to nonlinear dynamics: Continuous conductance based neural dynamics model*’). We reference these results in the main text in a new section of our Discussion, ‘*Interpretation of IV based LDS estimator applied to nonlinear dynamics*’ in addition to raising the many caveats the reviewer raises and discussing future extensions that may address them in our new Discussion section. We end this section with the following summarizing statement (line # 829):

In general, the form of dynamics in the fly brain even for our simplified conductance model is highly under constrained--the parameters needed to evaluate the Jacobian are not available (average voltage, membrane time constants, post-synaptic currents associated with different synapse types and morphologies, etc). It is beyond the scope of this work to choose a good model, parameter distributions, and characterize eigencircuits as a function of these. The effectome is a first order approximation and explicitly depends on the state of the nervous system, future work could focus on how sets of these first order estimates can be used to infer the appropriate models and parameters of nonlinear neural dynamics equations.

We believe our manuscript now provides greater insight into our estimator and its limitations but at the same time motivates concrete future extensions that can address these limitations.

Referee #3 (Remarks to the Author):

The paper by Pospisil, Aragon and Pillow presents theoretical considerations and results in the fruit fly related to the estimation of synaptic connectivity based on the effects of causal manipulations —the ‘effectome’—, and using the estimated connectome to derive functionally relevant circuits. The paper is well written —although at times the authors sacrifice some

nuance for flow and simplicity; specific comments below—, and the figures are informative. Scientifically, I find their use of the Instrumental variable (IV) approach interesting, and the Bayesian extension seems to significantly improve weight estimation during the simulation scenario they consider. Adding circuit priors to their IV models and identifying dominant ‘eigencircuits’ in the fly brain are sound steps that seem to yield interesting results. I have, however, a number of concerns that should be addressed:

1. The IV approach considers independent variables that affect both the outcome (variable Y in their Fig 1B) and dependent variables (X in their Fig 1B), in this case the activity of the ‘stimulated’ neurons and the observed neurons. The authors do that in their simulations by adding variable Z in Fig 1B, but they only consider the case in which this independent variable Z is shared between the outcome measurements (Y) and dependent variables (X). This is, of course, a very reasonable first step, but there are other scenarios that should be considered in order to establish the applicability of their approach in awake behaving animals. In both cases the question would be: do the inferred weights still accurately reflect the actual weights?

1a. In the most general case, the synaptic inputs to the stimulated (X) and observed (Y) subpopulations of neurons in a behaving fly will be different, since a specific behaviour will be associated with a specific activity pattern. This ongoing activity will in turn affect the observed responses in a behaviour-specific manner (e.g., Shelchkova et al Nature Comms 2023 showed in humans that M1 responses to S1 intracortical microstimulation are task-dependent). One way to model this scenario could be to provide two different independent variables Z_x and Z_y to the stimulated and observed neurons, and systematically manipulate their relationship (e.g., the strength of their correlation/coherence), characterising the accuracy of the estimated weights.

1b. Another related scenario to simulate is the case in which the independent variables are higher dimensional activity patterns (rather than a one dimensional signal), each of which is projected to the stimulated and observed neurons via different projection matrices. These projection matrices could span a ‘relationship space’ that goes between correlated to fully independent, and ideally, with different units receiving different levels of ‘noise’. This would mimic, to some extent, the recent Leifer lab paper (Randi et al Nature 2023) showing that anatomy-derived activity predictions in the worm perform fairly poorly if the influence of extrasynaptic communication is not considered (more on this on specific comment E below).

1c. Finally, it would be very compelling if the authors performed these simulations focusing on two functionally distinct pathways, such as the visual pathways and the motor/orientation pathways. Would there be a difference between the ability to recover the weights between functionally different structures? (I don’t feel this analysis is critical, but I do think that considering more complex independent inputs as outlined in comments 1a and 1b is).

The reviewer makes the important point that because of the nonlinear dynamics of biological neurons, effects between neurons can change depending on inputs (e.g., Shelchkova et al Nature Comms 2023). Our new analyses of the IV estimator in the context of a conductance

based neural dynamics simulation demonstrates exactly the reviewers point (SI Fig 3). We now emphasize the point that the effectome may vary based on the state of the nervous system (line # 778).

We find that the IV estimate converges to the Jacobian (matrix of partial derivatives w.r.t. voltage of each neuron) of the underlying neural dynamics equation, evaluated at the voltages of the neurons in the population (SI 'IV estimator applied to nonlinear dynamics: Continuous conductance based neural dynamics model', SI Fig 3). This result is entirely consistent with our notion of the effectome, the Jacobian captures the effect of a small perturbation of one neuron on any other neuron. Our linear simulation (ref{eq:discrete_linear_dynamics}) is a special case where the Jacobian is exactly the connectome. In the conductance model the effect of a perturbation varies with the state of the network and the Jacobian captures this notion because it varies with respect to neuronal voltage.

It is thus reasonable to question whether our estimator will converge given arbitrary distributions of inputs (including the specific simulations the reviewer recommends). We now clarify that given the assumptions of our derivation (line #886) our estimator will converge regardless of the input noise simulation.

We update:

The raw instrumental variables estimator we outline below requires no distributional assumptions on the random variables (L_t , ϵ_t).

To:

*The raw instrumental variables estimator we outline below requires no distributional assumptions on the random variables (L_t , ϵ_t). **Furthermore, the noise (ϵ_t) can take on any correlation structure over time and across neurons thus our estimator applied to dynamics of this form will converge to the true effectome, W regardless of confounding inputs (so long as the noise is independent of the stimulation, L_t). We note that for nonlinear dynamics further assumptions are required (see Discussion 'Interpretation of IV based LDS estimator applied to nonlinear dynamics').***

To concretely demonstrate this point we provide a simulation below where the degree of correlation in unobserved inputs to two neurons is varied (reviewer example 1a) and in all cases the IV estimator remains unbiased.

2. Figure 2: Perhaps I missed this in the Methods (which are too brief and should be significantly expanded), but the authors only seem to have validated their Bayesian IV approach on simulations in which they ‘stimulated’ a single neuron and recorded from the entire brain (Lines 220-21). However, later in the paper they argue that for practical and scientific reasons, stimulation should be applied to a specific population of neurons. Accordingly, the authors should validate their Bayesian IV approach when larger neural populations are simultaneously active.

We thank the reviewer for suggesting this important demonstration of the full capabilities of our method. To address this we have now run simulations showing that our Bayesian IV approach converges to the correct solution when larger neural populations are simultaneously active both in linear simulations (response SI Fig 5,6) and in our conductance based model (SI Fig 3,4). We now emphasize this in results (line # 358)

Our simulations thus far have focused on the stimulation of a single neuron while the entire fly brain is observed. Applying this approach to larger groups of neurons is straightforward (SI Fig \ref{fig_S3}-\ref{fig_S6}).

and methods (line # 947)

For clarity, in our example simulation we chose a single neuron to stimulate and estimate its downstream synaptic weights(Fig. \ref{fig3}). This neuron was chosen on the basis of having a larger than typical number of downstream contacts. It is straightforward to estimate downstream weights for multiple neurons simultaneously (eq. \ref{eq:solve_w_xy}) and we demonstrate this in both for IV and IV-Bayes (SI Fig \ref{fig_S5}) and IV in a conductance based model (SI Fig \ref{fig_S3}, \ref{fig_S4}).

We have also now significantly expanded our methods (an additional ~9 paragraphs) to more thoroughly detail our simulations.

3. As far as I could understand from the Methods, the authors always simulate brain dynamics as evolving according to a first order AR system. This is obviously the best first approximation, but I feel that the authors should simulate the effect of higher order dynamics on their estimations, since this would be a more realistic scenario. How would their ability to infer the

actual weights be affected by considering higher order autoregressive models? (I realise that inferring the weight matrix dynamics for the second order term may be tricky, but it could be some nonlinear function of the first order term matrix)

The reviewer raises an important point, our approach estimates effects at a single rapid time scale. It is thus important to understand how the estimator will behave in the context of neural dynamics where there are effects at multiple time scales. We address this by taking the reviewers suggestion and applying our estimator of AR(1) to a higher order AR(P) process. We hypothesized that because we showed our estimator was consistent regardless of the distribution of neural inputs that it would accurately estimate the first order component of an AR(P) process. We confirmed this hypothesis in simulation with our IV estimator applied to an AR(4) process with weights drawn from an IID standard normal then scaled for stability. We measure the accuracy of recovery of the first order effects (W_1) as a function of the number of time samples and find the IV estimator converges more rapidly if $P=1$ but still converges to the first order effects when $P=4$ (SI Fig 6).

SI Fig. 6: IV converges to first order effects in a higher order AR model. **(A) (left)** Example AR(1) IV estimate from data drawn from AR(4). **(A) (right)** Example AR(4) process weights drawn from an IID standard normal then scaled for stability. Note estimate and W_1 are nearly identical. **(B)** accuracy of recovery of the first order effects (W_1) as a function of the number of time samples.

We emphasize in the discussion an extension of the estimator to AR(P) will be an important line of future research (line # 712) so that effects at different time scales can be captured.

The instrumental variables approach was recently extended to discrete linear dynamical systems of arbitrary order (AR(P) with $P \geq 1$) ... An extension to AR(P) would relax

the restriction that the time scale of interaction is known and fixed (in our simulations 1 ms) thus slower effects (potentially through extra-synaptic paths) could be detected.

We thank the reviewer for suggesting this analysis that clarifies the interpretation of our estimator applied to processes with effects at multiple time scales.

4. The eigencircuit analysis suggests that the dynamics of the fly brain are quite high-dimensional. It could be interesting to see the effect of specific changes in circuit connectivity on this measure (as well as on the timescale analysis), since this could lead to specific predictions in certain lines. For example, what would happen if synaptic strength in specific regions is increased? Or if brain connectivity were less sparse? Or if regions were connected by more “hub” neurons?

We were also surprised at how our eigencircuit analysis indicated whole fly brain dynamics were high-dimensional—in contrast to many findings on the low dimensional nature of neural dynamics. We agree it would be interesting to better understand how this high-dimensionality arises by determining the perturbations that can change the dynamics. Cell-type and region-based analyses are excellent directions for future research that would require careful thought about the form of the relationship between cell-types and regions to effects between neurons.

To preliminarily address these questions we performed several global transformations of our original connectome and found the dimensionality result was quite robust to them (SI Fig 8). We applied a tanh nonlinearity and found dimensionality increased only slightly—even when only the sign of the connectome was preserved! Adding measurement error noise also did not change the dimensionality.

We considered whether the dimensionality might be a simple consequence of the high sparsity of the connectome. To test this we shuffled the synaptic weights of the connectome, corrupting patterns of connectivity while preserving the same marginal sparsity and found that the dimensionality was greatly increased (dotted lines). This suggests that dimensionality is less likely to be influenced by any global changes but instead by specific alterations to patterns of connectivity as suggested by the reviewer.

We now include these results in our supplement (SI Fig 8) along with the inclusion of the reviewers idea to study dimensionality to motivate future work in this promising direction (line # 1787):

We wondered how this low-dimensionality related to connectivity. Cell-type and region-based analyses are excellent directions for future research that would require careful thought about the form of the relationship between cell-types and regions to effects between neurons.

SI Fig. 8: Effect of nonlinearity and measurement error applied to connectome on dimensionality. For reference the original eigenvalues of the connectome (scaled by the largest magnitude eigenvalue) are plotted (black). Non-linearity of increasing degree via hyperbolic tangent applied to connectome weights scaled relative to their maximum (larger scaling, stronger effect of nonlinearity) we found a small increase in dimensionality (pink, orange, red above black). In an extremal case we set the connectome weights to their sign so that all entries were either +1, 0, or -1 and found dimensionality was similar (cyan overlaps black). Shuffle control where index of count entries was shuffled without replacement to determine if dimensionality was result of marginal connectome statistics (e.g., sparsity). We found large increase in dimensionality (n=5 independent shuffles, dotted lines well above black). Simulated measurement error with draws from Poisson distributions with means equal to the original synapse count (n=5 simulations, dashed line).

4.1. Relatedly, I found it surprising that the first eigencircuits have very fast dynamics and involve very few neurons. The approach the authors took to arrive to this conclusion is reasonable, but what would happen if they increased the percentage of eigenvector loading power from 75 % to, e.g., 90 % or even 95 %, would their results change very significantly?

We were also surprised that the early eigencircuits involved so few neurons and had the maximal possible variation in time. To gain further insight into the former we plotted the number of neurons involved in each eigencircuit as a function of increasing eigenvector loading power and found that the trend in neuron numbers is consistent across eigenvector loading powers between 75% and 95%. These results are shown in SI Fig 17B.

SI Fig. 17: Robustness of non-localized eigencircuits to choices in loading threshold. **(A)** Localization index computed across different inclusion criteria. **(B)** Number of neurons within each eigencircuit for multiple loading thresholds.

With respect to the latter we now clarify in the text that the time scale of the all eigencircuits (and all results in Fig 4A-E) were calculated from the entire connectome (line # 503).

We updated:

For each example circuit, we visualized the neurons that together comprise 75% of loading power within their respective eigenvector.

To:

*For each example circuit, we visualized the neurons that together comprise 75% of loading power within their respective eigenvector (**in contrast all results in Fig ref{fig3}A-E are for all neurons**).*

4.2. Crucially, the authors should investigate whether the inferred eigencircuits are consistent across different sets of hyper parameters (at least the loading power mentioned above)? A sensitivity analysis should be performed.

We agree that the robustness of eigencircuits is an important question: would our results change if the protocol for measuring the connectome had changed, or if the connectome of a different fly had been measured? We addressed this by sampling noisy connectomes to model both biological variability (across animals) and measurement error (missed or miscounted synapse numbers). We do so by computing the eigencircuits of the connectome plus gaussian noise proportional to the synaptic count (a linear mean variance relationship, common in counting distributions). We then measured all pairwise correlations between the top 250 eigencircuits between the original and corrupted connectome (removing redundant conjugate pairs). We then plotted the max absolute r-value for each original eigencircuit. For the eigencircuits with only real parts this was a straightforward calculation of r-values. For the

complex eigencircuits (with real and imaginary coefficients) we regressed the real and imaginary components onto those of another then calculated the r-value of this linear fit. We find that in general for higher SNR 10-100 the top eigencircuits are nearly identical (black, grey traces near 1). For lower SNR's most of the top eigenvectors are surprisingly still recovered but a few are lost (red trace below $|r| \sim 0.5$).

SI Fig. 9: Robustness of eigencircuits to measurement error. Maximum absolute correlation between each original eigencircuit and all eigencircuits from connectome with gaussian noise added. For complex eigenvectors we regressed the real and imaginary components onto those of another then calculated the r-value of this two parameter linear fit. Measure was taken for three levels of noise (SNR=100 $\sigma^2 = 0.01\mu$ black, SNR=10 $\sigma^2 = 0.1\mu$ grey, and SNR=1 $\sigma^2 = \mu$ red where μ is the synapse count).

Thus overall early eigencircuits appear quite robust, we now include these results in our new supplement section 'Additional connectome analyses' (line # 1730) and reference them in our results section (line # 524).

To preliminarily address non-localized eigencircuits we considered whether modest amounts of biological variability, measurement error or saturation would change these results (SI Fig \ref{fig_S8}-\ref{fig_S11}) and found that in general the early eigencircuits were quite robust and later less so. Interestingly we found the non-local circuits tended to be less robust but there were many examples that were very robust to perturbation (SI Fig \ref{fig_S10}). We found anatomically localized eigencircuits were in the minority (~10 \% of the top 1000 eigencircuits) and non-localized circuits were often among the most dominant (SI Fig \ref{fig_S7} E,F).

We thank the reviewer for motivating this analysis as it strengthens the eigencircuit results of the paper by showing they are robust.

4.3. The ‘not anatomically localised circuits’ merit further investigation. In particular, the authors should verify how robust these circuits are to different parameter choices. Moreover, I think that their proportion could perhaps be used as a goal to be minimised—I’m not implying that the authors should develop an optimisation process to minimise this quantity, but rather that perhaps this number exhibits some relationship with specific parameters that could reveal an interesting trend.

We agree with the reviewer, the non-local eigencircuits merit further investigation—especially given the bulk of the fly community has focused on studying local circuits. Thus to the degree we can verify these circuits are meaningful we may motivate studies of novel global circuits discovered through a data-driven approach.

We address this by first developing a ‘locality’ index: the fraction of synapses formed in a single neuropil for pairs of neurons in the top 75% of eigencircuit loading. We then studied this quantity as a function of eigencircuit magnitude where we found that lower modes tended to be less local but there were many exceptions to this tendency (i.e., dominating non-local eigencircuits; SI Fig 7 E,F). Overall we found that local eigencircuits were actually rare (~10%). Below we consider whether the pre-preponderance of non-local circuits may be the result of noise—as we would expect random weights would tend to give non-local circuits.

To determine non-local eigencircuits robustness we examined the correlation between this localization index with the degree of corruption from the highest noise added to the connectome condition above (red). We found a significant correlation where more localized circuits were more robust to added noise. Yet there were many examples of non-localized circuits that were robust to noise.

SI Fig. 10: Non anatomically localized eigencircuits tend to be less robust. Robustness of eigencircuits measured as maximum absolute correlation with eigencircuits from noisy connectome (SNR=1, see SI Fig 9 red trace) plotted against locality index, fraction synapses in one neuropil for top 75 % of eigenvector loading of each eigencircuit.

We thus conclude that in some cases the non-local eigencircuits we found may be sensitive to large amounts of measurement error but that there were many robust non-local eigencircuits. We now provide these results in the supplement and reference them in the results section (line # 524):

To preliminarily address non-localized eigencircuits we considered whether modest amounts of biological variability, measurement error or saturation would change these results (SI Fig 8-11) and found that in general the early eigencircuits were quite robust and later less so. Interestingly we found the non-local circuits tended to be less robust but there many examples that were very robust to perturbation (SI Fig 10). We found anatomically localized eigencircuits were in the minority (~10 % of the top 1000 eigencircuits) and non-localized circuits were often among the most dominant (SI Fig 7 E,F).

Overall we believe these results further motivate future investigation of the non-local eigencircuits we have discovered by showing they are often quite robust.

5. The eigencircuit idea is very interesting, and deserves further investigation:

5.1. Providing a more quantitative description of their findings including the distribution of timescales, number of neurons, (spatial) distance between neurons, distribution across areas, etc., would be very interesting. This should be done for a reasonable number of eigencircuits, in addition to the two that are discussed in depth in Fig 5.

We thank the reviewer for highlighting the concept of eigencircuits as interesting and deserving of further investigation. Their suggested analyses led to several new insights we summarize in the introduction of this document and include in our supplement and reference in our results. Below we quote from the supplement where we detail our new analyses

We focused our additional analyses on the unique eigencircuits associated with the first 1000 eigenvalues (516 because complex eigenvalues come in conjugate pairs). We analyzed the distribution of time scales, localization of synapses, synapse count, and sign. We find that the fastest time scales tend to be disproportionately in the earliest modes and slower time scales in lower modes (SI Fig 7). We interpret this to imply that the largest dynamical modes in the fly reflect fast reciprocal inhibition instantiated with large numbers of synapses. Most eigencircuits have rotational dynamics at intermediate time scales (B). In general eigencircuits tend to have similar numbers of excitatory and inhibitory synapses but when there is an imbalance synapses tend to be predominantly inhibitory (C). There was a tendency for the fastest time scales to have majority inhibitory weights (D). Most eigencircuits were not localized but localized circuits could be found up to the 200th eigencircuit (E,F) and there were certainly dominating non-local circuits (e.g., eigencircuits 2, 6, 7).

SI Fig. 7: Summary statistics of time scales, locality, and synapse properties. **(A)** Angle in complex plane of eigenvalues associated with eigencircuits plotted as function of eigencircuit rank (w.r.t., eigenvalue magnitude). The angle 0 implies the slowest possible dynamics where eigencircuit monotonically decays. The angle 180 implies fastest possible dynamics where the sign of the eigencircuit flips at every discrete time step. **(B)** Histogram of angles. **(C)** Average synapse count of excitatory neurons that form synapses in top 75 % of eigenvector loading (red), average for inhibitory (blue) and across both (black). **(D)** Plot of the fraction excitatory synapses of all synapses formed in top 75 % of eigenvector loading of each eigencircuit as a function of time scale (eigenvalue angle). **(E)** Locality index, fraction synapses in one neuropil top 75 % of eigenvector loading of each eigencircuit. **(F)** Cumulative distribution of (E)

In the results of Fig 4 we now include the new property we measure for our example neurons (line # 550).

We found that the first eigenvector localized to the lobula plate (Fig \ref{fig4}A) and was highly sparse (Fig \ref{fig4}B). We quantified the anatomical localization as the fraction synapses in a single neuropil. For this eigencircuit 75 % of synapses were in a single neuropil the lobula plate the other were in the inferior posterior slope.

(line # 631)

We found that eigenvector 42 contained high loadings for GABAergic R4d ring neurons in the ellipsoid body (Figure \ref{fig4}G). For this eigencircuit 99 % of synapses were in a single neuropil the ellipsoid body, the others were in the fan-shaped body and mushroom body medial lobe.

... loadings that accounted for 75 % of all eigenvector power. In the case of the opponent motion circuit this was 5 neurons and in the case of the ellipsoid body circuit this was 21 neurons.

Overall we believe these further characterization of eigencircuits (in addition to those in Fig 3) strengthen the manuscript by providing global insights into the connectome of the fly while also motivating and guiding future work to examine these putative circuits in detail. For example circuits that are robust and representative of interesting variation across circuits (local vs non-local, slow vs fast, etc) as indicated by our analyses will be far more compelling to examine.

5.2. My understanding is that the authors performed this analysis under the assumption of linear summation of synaptic weights. This is a reasonable first order approximation but the authors should check whether nonlinear functions (e.g., an exponential) allow recovering similar eigencircuits. This feels especially important given both the account of nonlinear summation by some neuron types, and that they use linear methods for matrix factorisation to identify the eigencircuits.

We appreciate the reviewer's suggestion to consider nonlinear synaptic integration—a known property of biologically realistic neurons. We make a first order attempt to address the reviewers request by considering a nonlinear transform of synaptic weights (see below for a more thorough discussion). We used a saturating function (hyperbolic tangent) following the intuition that the magnitude of effects between neurons will eventually saturate as postsynaptic voltage nears the synaptic reversal potential (a fact not accounted for by classic firing rate models of neural dynamics). We titrated the magnitude of this effect by rescaling the connectome weights relative to the max count of synapses (SI Fig 11 A). In general we found a gradual degradation of the original eigencircuits. Surprisingly, some eigencircuits were even retained after only the sign of the connectome was preserved (B green trace above 0.5).

SI Fig. 11: Robustness of eigencircuits to nonlinearity applied to connectome. **(A)** Example plot of hyperbolic tangent transformation (tanh) of connectome weights scaled by half the max synapse count ($c=2$). **(B)** Plot of maximum absolute correlation of original vs transformed connectomes (see SI Fig 9 legend) versus rank of original eigencircuit, split across real and complex unique eigencircuits of top 250 eigencircuits.

We note that the eigendecomposition is an explicitly linear decomposition thus now clarify in the text that these eigencircuits are an approximation to nonlinear dynamics including nonlinear synaptic integration in our new Discussion section, ‘Interpretation of IV based LDS estimator applied to nonlinear dynamics’ (line # 775).

To provide a concrete interpretation of linear model estimates applied to nonlinear neural dynamics we analyzed our estimator in the context of a conductance-based model of neural dynamics. This model includes both a spiking nonlinearity and nonlinear synaptic integration.

With the new connection we have formed between the Jacobian and our estimator applied to a conductance model we can make the connection to nonlinear synaptic integration concrete. In a conductance model synaptic integration is multiplicative with the voltage of the integrating cell (SI eq 3). Thus nonlinear synaptic integration is reflected in the effectome of a conductance model via each row being scaled by the average voltage of the neuron associated with that row. Measuring the effectome at several voltage levels could reveal this tell-tale sign of synaptic integration following a conductance model.

The connection of the effectome to the Jacobian of conductance model also provides a concrete interpretation of the signed synaptic counts as a linear dynamics matrix which we now include in our discussion (line # 819)

It is only in a narrow set of situations in which the best linear approximation to these nonlinear neural dynamics would be proportional to the synaptic count and sign. Variation in neural voltage across neurons, synaptic reversal potentials, synaptic conductances, membrane time constants, and more can all corrupt proportionality (SI eq 3). Thus a biophysical interpretation of our results is that the eigencircuits decompose the best linear approximation to neural dynamics under the assumption there is a 'small' amount of variation across these parameters. A mild confirmation of this untested assumption is that we do in fact recover known functional sub-circuits of the fly nervous system (Fig 4).

We further caution that (line # 829):

In general, the form of dynamics in the fly brain even for our simplified conductance model is highly under constrained--the parameters needed to evaluate the Jacobian are not available (average voltage, membrane time constants, post-synaptic currents associated with different synapse types and morphologies, etc). It is beyond the scope of this work to choose a good model, parameter distributions, and characterize eigencircuits as a function of these. The effectome is a first order approximation and explicitly depends on the state of the nervous system, future work could focus on how sets of these first order estimates can be used to infer the appropriate models and parameters of nonlinear neural dynamics equations.

We thank the reviewer for pushing us to consider more biologically realistic scenarios, we feel that the new analyses and text we have included to address their concerns both clarify the estimator and eigencircuits of the connectome and consequently make them more compelling as objects of future research.

6. The paper is well-written, but at times I struggled with the overarching logical flow. Could the authors confirm that my understanding is correct (and edit the manuscript for high-level clarity if they feel it's appropriate)? First, the authors show that they can estimate the connection weights of a fly using single cell stimulation and whole brain recordings provided that they add prior knowledge on fruit fly brain connectivity (Figs 1 and 2).

The reviewer is correct. Though the prior is not strictly necessary but drastically increases the efficiency of the estimation—making the approach more experimentally feasible.

Then they argue that stimulation of a subpopulation of neurons together with partial brain recordings is the best approach in practice (Fig 3) —a section that could be shortened, and that again calls for an extension of their simulation to this type of scenarios as I commented on #1 above.

We have now included multi-neuron stimulation simulations in a conductance model (SI Fig 3,4) and in linear simulation (SI Fig 5,6). We now reference these simulations in our results (line #

358):

Our simulations thus far have focused on the stimulation of a single neuron while the entire fly brain is observed. Applying this approach to larger groups of neurons is straightforward (SI Fig 3-6).

Finally, they show that eigendecomposition of the 'measured' connectome can help identify specific 'eigencircuits' and form hypotheses about their computations (Fig 4); the authors provide two such examples (Fig 5). If this reading is correct, to me, the initial emphasis on the effectome detracts from the fact that it's a means to estimating the connectome, which is the main object whose analysis can reveal interesting circuits and their computations.

We thank the reviewer for their close reading of our manuscript and their suggestions to help clarify the overall flow of the manuscript. In our simulations our estimator does in fact recover the ground truth connectome (perturbed by Poisson-like variability to model biological variability) yet this is not the purpose of the estimator. Our estimator was designed to infer the sign and magnitude (if any) of direct (monosynaptic) effect between neurons (we term this the effectome). It is plausible that these effects will on average be proportional to the number of synapses between neurons (thus why we use the connectome as a prior) but it is also highly plausible they will not be. We discuss this in our new supplementary section '*IV estimator applied to nonlinear dynamics: Continuous conductance based neural dynamics model*'. We show that for a conductance based model when there is a large variation in average firing rate across neurons the effectome will have a weak relationship to synaptic counts (as a proxy for conductances). We thank the reviewer pointing out that as written the manuscript is confusing on this point thus we now explicitly clarify this point early on in the results (line # 336).

We have updated :

This corruption could reflect natural variation between flies' connectomes or measurement error in the connectome.

To (line # 336):

*This corruption could reflect natural variation between flies' connectomes or measurement error in the connectome. **We note that the IV estimator is not meant to estimate the underlying connectome but instead local linear effects between neurons which might have a weak relationship with the connectome that could even vary depending on the state of the nervous system (see Discussion, 'Interpretation of IV based LDS estimator applied to nonlinear dynamics').***

Nonetheless we analyze the connectome, as if it were the effectome, because it is a simple, plausible first guess at the effectome thus can provide guidance about which parts of the effectome to focus on estimating. See our response to reviewer 3 section 5.2 for a more thorough justification but roughly speaking by using the connectome as a stand-in for the effectome we implicitly assume the variation in model parameter across neurons is minimal

(e.g., average voltage, firing rate nonlinearities, membrane time constants, etc). We now clarify this point in our discussion (line # 823):

Thus a biophysical interpretation of our results is that the eigencircuits decompose the best linear approximation to neural dynamics under the assumption there is a 'small' amount of variation across these parameters. A mild confirmation of this untested assumption is that we do in fact recover known functional sub-circuits of the fly nervous system (Fig 4).

7. The authors should provide more details on the Methods; I struggled finding a number of details (e.g., how were the visual inputs in Fig 5 simulated?).

We have now expanded our section 'Simulations to analyze eigencircuits' (line # 901), including additional detail on both simulations (line # 913):

The global scale of W was hand tuned to recapitulate the qualitative findings of prior work. It is unknown what a good approximate scaling of synapse number to linear effect is in these regions but if the scaling was zero then this circuit would not perform a computation and if it was too large the dynamics would be unstable. Future work could consider other normative motivations for scaling these weights (e.g., \cite{lappalainen_connectome-constrained_2023}).

In both the ellipsoid and opponent motion simulations inputs in Fig 4 were simulated by adding in a step function input to the relevant neurons in our dynamics simulation. In both cases stimulation, the 'step', lasted 30 ms. This step function was either at 0 or a hand tuned maximum.

In the opponent motion circuit (Fig 4 A) input maximum was 0.001. In the unilateral back-to-front motion case only VCH and DCH neurons receive input. In the condition of BTF motion to both eyes we add in the same input to VCH, DCH, LPi2-1, and Am1. For the ellipsoid body circuit (Fig 4 G) the 'winner' neuron receiving a larger input had input of 0.0015 and other neurons had 0.0009. In both cases stimulation lasted 30 ms. In the case of the opponent motion circuit we used prior literature to determine which neurons to stimulate under unilateral and bilateral back-to-front motion \cite{shinomiya_neuronal_2022, haag_recurrent_2001}. In the case of the ellipsoid body circuit the ordering of the neurons with respect to retinotopic input was arbitrarily set by their eigenvector magnitude.

We also link to simulation code that exhaustively detail our methods (line # 1068)

Code availability: all code to analyze data and generate figures is available at <https://github.com/dp4846/effectome>

We also clarify the reasoning behind the patterns of inputs to the opponent motion circuit in the text (line # 609)

It is known that VCH and DCH receive major input from H2 from the contralateral eye, which responds to back-to-front (BTF) motion (Haag & Borst, 2001), whereas, LPi2-1 and Am1 receive major inputs from T4b/T5b which are driven by ipsilateral BTF motion (Shinomiya et. al., 2022).

We have also now significantly expanded our methods more generally (an additional ~9 paragraphs) to more thoroughly detail our simulations.

SPECIFIC COMMENTS

A. Lines 63-64: the authors claim that optogenetics likely meet the criteria for being an IV but, in my view, this may only apply to very brief stimulation pulses. Long stimulation trains may put the system in 'unnatural neural states', potentially altering the 'natural' mapping between the stimuli and effects via complex changes in independent factors (e.g., saturation of certain cells, etc) that simple models won't be able to capture.

We agree with the reviewer, in general there is a tension between increasing the strength of stimulation and keeping the nervous system in a 'natural' state. In practice the experimentalist would want to titrate stimulation to the minimal amount possible and test whether results changed significantly with different degrees of stimulation. We now include this important point in discussion (line # 785)

In the conductance model the effect of a perturbation varies with the state of the network and the Jacobian captures this notion because it varies with respect to neuronal voltage. This also provides a more precise motivation to titrate optogenetic stimulation to the minimal amount possible---for larger perturbation the estimate of the Jacobian becomes less precise and averages across more states.

B. The authors explain in the Introduction that they are interested in the neural interactions that contribute most strongly to whole-brain dynamics (Lines 94-95). This seems very sensible —and is a more achievable aim—, but a variety of lines of evidence indicate that the largest components in neural activity may not be directly related to the production of behaviour, at least in mammals (e.g., the observation that activity in mouse V1 is mostly related to arousal/behavioural state in Stringer et al Nature 2019, the observation that most of the variance in monkey PFC during sensory-based decision making is not related to either the stimulus or the decision [Kobak ... Machens et al eLife 2016], the finding that monkey M1 is dominated by signals that do not relate to the produced movement [Churchland lab]). The question then becomes, can the method in this paper potentially allow us to relate specific activity patterns and behaviours/sensations or is it limited to much coarser patterns? (I am not challenging the authors' core hypothesis, just pointing this out in case they want to incorporate these ideas into the paper).

We thank the reviewer for pointing out connections to the broader systems neuroscience literature. We now include these ideas in our discussion (line # 794):

More generally extensions to our estimator can employ estimated trajectories within repeated experimental conditions, stereotyped behavior, or inferred latent states \cite{linderman_bayesian_2017}. The latter may be critical if the emerging observation that the bulk of variation in global brain dynamics is unrelated to stimuli or behavior holds in the fly \cite{russo_motor_2018, stringer_spontaneous_2019, kobak_demixed_2016}.

C. This is not my paper and as such this is just a suggestion, but I felt that the relatively long section around Fig 3 sidetracked my thoughts when reading the paper. I don't have a specific suggestion but perhaps the authors could consider reformatting? Leaving the key idea in the discussion and appendix?

This is an excellent suggestion that improves the flow and clarity of our manuscript. We have moved this figure and section to the supplement.

D. Figure 5D,J: The synaptic count x sign scalebar always takes very negative values, so the chosen color scale is odd. Relatedly, is this the case across all dominant modes?

We made the scale bar symmetric as we wanted to emphasize that all the weights were inhibitory. For eigenvectors with negative real eigenvalues (the case for both these examples) their synaptic weights tended to be negative. For real positive eigenvalues synaptic weights tended to be uniformly positive and for imaginary eigenvalues they were mixed. See SI Fig 7 C.

E. The paper is focused on predicting neural responses from optogenetic stimulation to build a functional map of the fly brain. While it's a very interesting topic, I think that philosophically speaking a 'causal model of the brain' should be about predicting behavioural output, not just neural responses. This should be recognised in the paper.

We agree with this point and we now indicate in the discussion that future work should extend these models by incorporating known relationships between motor neurons and behavior (line # 767).

To link estimated internal dynamics models to behavior, a central goal of systems neuroscience, will require integration with models of how descending motor neurons actuate the body \cite{roman_whole-body_2024, braun_descending_2024, karaschuk_sensorimotor_2024}.

Thus by learning the effect of internal dynamics on motor neurons we can then by extension learn how behavior is generated.

F. Neuropeptides: Recent work in the worm from the Leifer lab (Randi et al Nature 2023) shows that non-synaptic communication (through neuropeptides) may be a large contributor to the generation of neural activity, helping to explain as much as 30 % of the total "neural variance". The authors mention their potential role as shared (between the responses of observed neurons and that of the unobserved 'outcome neurons') independent inputs, but the Leifer paper

suggests that their contributions may be much higher dimensional. This should at least be discussed, and some claims in the paper softened, e.g., Lines 214-15 “Thus IV is not corrupted by unknown, unobserved inputs”. The authors should also model this scenario as I indicate in comment #1 above.

We thank the reviewer for making this connection with more recent literature and this potential mismatch between chemical synapses and effects between neurons. We have now added additional simulations relevant to extra-synaptic effects such as neuropeptides including a demonstration that our estimator could discover extra-synaptic effects missed by the connectome (SI Fig 5) and consideration of effects at slower time scales (SI Fig 6).

We use these observations to help motivate research on extensions to IV-Bayes to multiple time scales (VAR(P)) (line # 713)

An extension to VAR(P) would relax the restriction that the time scale of interaction is known and fixed (in our simulations 1 ms) thus slower effects (potentially through extra-synaptic paths such as peptide signaling pathways) could be detected.

We have also qualified the claim originally on line 214-215 now on line # 246

“Thus IV is not corrupted by unknown, unobserved inputs”

To

“Thus IV, **under our assumed model**, is not corrupted by unknown, unobserved inputs”

Reviewer Reports on the First Revision:

Referees' comments:

Referee #1 (Remarks to the Author):

The authors did a good job addressing my concerns about robustness to errors in the connectome prior and I am convinced that IV-bayes would do a good job learning interactions from real data. The new connection between the IV-Bayes solution and the Jacobian of the neural dynamical system is a wonderful addition and also helps allay my concerns about generality of the IV-Bayes solution.

I still have concerns about the feasibility of the experimental approach - in particular the sections in the SI describing how to measure the whole brain effectome. In my opinion the focus should be on learning the effectome for small, tractable subnetworks. The whole-brain effectome approach is frankly unrealistic and may prevent experimentalist readers from taking the proposal seriously. But, ultimately, I think they have done enough by extending the description of how to explore subnetworks. I leave further changes in this direction to the authors discretion and in either case am comfortable with the description of the proposed experimental plans as written.

Referee #1 (Remarks on code availability):

The code runs fine. It has not been updated to reflect the revisions in the current version of the manuscript and should be updated before publication.

Referee #2 (Remarks to the Author):

Thanks for the authors' detailed responses and adding extensive new results to address my concerns.

I appreciate authors' efforts to demonstrate the methods to be applicable in more biological realistic scenarios where the nonlinearity and nonstationarity play a big role. The authors extend their methods and evaluate on nonlinear dynamical system with a conductance based model, and shows convergences to Jacobian matrix of the dynamical equations. As well as applying switching linear dynamical system for non-stationary dynamics.

Meanwhile, the extended results with more samples for eigencircuits and discussions on nonlocal circuits are valuable.

Suggestions on including related works on connectome constrained modeling are well received.

The only concerns left is that most of the studies are based on simulations instead of real data. I agree with reviewer 1 that the paper is more like a proposal which provides a roadmap to the future studies instead of delivering more concrete findings, though it could still be valuable. Meanwhile, the first part for instrumental variables with optical inputs is a bit detached with the second part to

discover eigencircuits.

Referee #2 (Remarks on code availability):

I think the code provides basic functions to generate the figures in the paper. While it failed to provide enough instructions for downloading data, installing the environment, and provide guidance to run the applications. It is hard for their developed tools and models, simulators to be reused by the research community. I didn't manage to install and run the code.

Referee #3 (Remarks to the Author):

I wanted to thank the authors for their detailed responses to my previous comments, and the addition of the missing details to the methods (and for sharing the code). I believe that the manuscript has improved substantially after peer review, for which I would like to congratulate the authors. I only have a few minor additional suggestions and comments:

- I think some references to supplementary figures may be wrong (e.g., bottom of page 33 refers to Fig S2D but should be Fig S3D; same for bottom of page 34, the text refers to Fig S3 but convergence speed is shown in Fig S4C). In addition, Fig S6 seems never to be cited in the text or discussed in the supplement.

- Lines 1605-7 read: "More broadly, conditioning the IV estimate on randomized experimental interventions or external covariates (e.g., behavior) that are independent of optogenetic perturbation will reduce the variance of neural dynamics (i.e., $\text{Var}[Y] \geq E[\text{Var}[Y | X]]$)." -> Optogenetic stimulation should elicit responses that are constrained by circuit connectivity (in terms of the observed covariation across neurons) and these similar constraints should apply to behaviour (I am thinking about the preservation of ring and toroidal manifolds between wakefulness and sleep in mouse thalamus and MEC, respectively, and the preservation of population dynamics between visual and optogenetic stimulation in mouse V1 showed by the Deisseroth lab in Marshel et al Science 2019). It then follows that perhaps the same covariance (or at least similar) will be observed during behaviour and optogenetic stimulation? (If the authors agree, I'd encourage them to rephrase these sentences)

- On the high-dimensionality of neural dynamics: Like the authors, I was surprised by the finding that the linear embedding dimensionality of the connectome (borrowing the term from Ostojic and Jazayeri Curr Opin Neurobiology 2023) of the fly brain being relatively high. Regarding this finding, I'd like to first comment that I really like the new analyses in Figure S8. Second, I was wondering whether this apparently high dimensionality may become lower during behaviour, since behaviour (and probably just having a body?) provides additional constraints. What do the authors think about this? Do they have a way to test this with their current model, at least in terms of providing stimuli? If not, it may be worth adding a few lines to this regard to the manuscript, especially since the dimensionality of neural activity is a hot discussion topic.

Referee #3 (Remarks on code availability):

Unfortunately I haven't had the time to review the code, I did check out the repo and the output figures though.

Author Rebuttals to First Revision:

We thank the reviewers for their thorough reading of our revised manuscript and kind comments pertaining to our revisions. Below, we provide a point-by-point response to the remaining concerns raised by each reviewer (in blue text), with our response and changes given below (in black and black italics, respectively). We believe the manuscript is stronger, and hope that the reviewers will be persuaded that it is now suitable for publication.

Referee #1 (Remarks to the Author):

The authors did a good job addressing my concerns about robustness to errors in the connectome prior and I am convinced that IV-bayes would do a good job learning interactions from real data. The new connection between the IV-Bayes solution and the Jacobian of the neural dynamical system is a wonderful addition and also helps allay my concerns about generality of the IV-Bayes solution.

I still have concerns about the feasibility of the experimental approach - in particular the sections in the SI describing how to measure the whole brain effectome. In my opinion the focus should be on learning the effectome for small, tractable subnetworks. The whole-brain effectome approach is frankly unrealistic and may prevent experimentalist readers from taking the proposal seriously. But, ultimately, I think they have done enough by extending the description of how to explore subnetworks. I leave further changes in this direction to the authors discretion and in either case am comfortable with the description of the proposed experimental plans as written.

We greatly appreciate the skepticism of the reviewer. We acknowledge that the approach is not feasible in the near term future. We emphasize that integrating all of the techniques we propose into the fly will be challenging and will require a concerted effort to achieve. We now reference our concrete proposal for a small tractable eigencircuit for future study via this approach

*In the supplement we discuss how to achieve this ideal setting **including practical steps for generating a fly line specific to estimating an eigencircuit (see SI Experimental approach and SI Sparse expression in selected population respectively)***

To further emphasize the prospective nature of our paper we have updated our title from:

From connectome to effectome: learning the causal interaction map of the fly brain

To:

The fly connectome reveals a path to the effectome

Referee #1 (Remarks on code availability):

The code runs fine. It has not been updated to reflect the revisions in the current version of the manuscript and should be updated before publication.

Our code is now updated to the most recent revision.

Referee #2 (Remarks to the Author):

Thanks for the authors' detailed responses and adding extensive new results to address my concerns.

I appreciate authors' efforts to demonstrate the methods to be applicable in more biological realistic scenarios where the nonlinearity and nonstationarity play a big role. The authors extend their methods and evaluate on nonlinear dynamical system with a conductance based model, and shows convergences to Jacobian matrix of the dynamical equations. As well as applying switching linear dynamical system for non-stationary dynamics.

Meanwhile, the extended results with more samples for eigencircuits and discussions on nonlocal circuits are valuable.

Suggestions on including related works on connectome constrained modeling are well received.

The only concerns left is that most of the studies are based on simulations instead of real data. I agree with reviewer 1 that the paper is more like a proposal which provides a roadmap to the future studies instead of delivering more concrete findings, though it could still be valuable. Meanwhile, the first part for instrumental variables with optical inputs is a bit detached with the second part to discover eigencircuits.

We totally agree with the reviewer that ideally this paper would include studies based on real data. Yet for the approach that we propose the integration of all the techniques we propose into the fly is unlikely to occur in the near term without a concerted effort and significant resources. We now emphasize in the discussion that our goal with this study is to demonstrate the approach is statistically feasible given the structure of the connectome in order to motivate such a concerted effort.

To further emphasize this point we have updated our title from:

From connectome to effectome: learning the causal interaction map of the fly brain

To:

The fly connectome reveals a path to the effectome

Referee #2 (Remarks on code availability):

I think the code provides basic functions to generate the figures in the paper. While it failed to provide enough instructions for downloading data, installing the environment, and provide guidance to run the applications. It is hard for their developed tools and models, simulators to be reused by the research community. I didn't manage to install and run the code.

We thank the reviewers for their careful evaluation of our code. We will improve the readme to better detail the entire process of running the code.

Referee #3 (Remarks to the Author):

I wanted to thank the authors for their detailed responses to my previous comments, and the addition of the missing details to the methods (and for sharing the code). I believe that the manuscript has improved substantially after peer review, for which I would like to congratulate the authors. I only have a few minor additional suggestions and comments:

- I think some references to supplementary figures may be wrong (e.g., bottom of page 33 refers to Fig S2D but should be Fig S3D; same for bottom of page 34, the text refers to Fig S3 but convergence speed is shown in Fig S4C). In addition, Fig S6 seems never to be cited in the text or discussed in the supplement.

We thank the reviewer for their careful reading of the manuscript and have now ensured that all figures are referenced and referenced correctly.

- Lines 1605-7 read: "More broadly, conditioning the IV estimate on randomized experimental interventions or external covariates (e.g., behavior) that are independent of optogenetic perturbation will reduce the variance of neural dynamics (i.e., $\text{Var}[Y] \geq E[\text{Var}[Y|X]]$)." -> Optogenetic stimulation should elicit responses that are constrained by circuit connectivity (in terms of the observed covariation across neurons) and these similar constraints should apply to behaviour (I am thinking about the preservation of ring and toroidal manifolds between wakefulness and sleep in mouse thalamus and MEC, respectively, and the preservation of population dynamics between visual and optogenetic stimulation in mouse V1 showed by the Deisseroth lab in Marshel et al Science 2019). It then follows that perhaps the same covariance (or at least similar) will be observed during behaviour and optogenetic stimulation? (If the authors agree, I'd encourage them to rephrase these sentences)

We agree with the reviewer that it is certainly possible covariance will remain similar during optogenetic stimulation. Indeed this would be the ideal condition for estimating the effectome. We apologize for any confusion we have now rewritten this sentence to better emphasize our goal is reduce the conditional variance i.e. for estimating the local dynamics around an average voltage ideally the variance about the average is minimal:

We now update:

More broadly, conditioning the IV estimate on randomized experimental interventions or external covariates (e.g., behavior) that are independent of optogenetic perturbation will reduce the variance of neural dynamics (i.e., $\text{Var}[Y] \geq E[\text{Var}[Y|X]]$).

To:

*More broadly, conditioning the IV estimate on randomized experimental interventions or external covariates (e.g., behavior) that are independent of optogenetic perturbation will reduce the **conditional** variance of neural dynamics (i.e., $\text{Var}[Y] \geq E[\text{Var}[Y|X]]$).*

- On the high-dimensionality of neural dynamics: Like the authors, I was surprised by the finding that the linear embedding dimensionality of the connectome (borrowing the term from Ostoic and Jazayeri Curr Opin Neurobiology 2023) of the fly brain being relatively high. Regarding this finding, I'd like to first comment that I really like the new analyses in Figure S8. Second, I was wondering whether this apparently high dimensionality may become lower during behaviour, since behaviour (and probably just having a body?) provides additional constraints. What do the authors think about this? Do they have a way to test this with their current model, at least in terms of providing stimuli? If not, it may be worth adding a few lines to this regard to the manuscript, especially since the dimensionality of neural activity is a hot discussion topic.

We completely agree with the reviewer that there are additional constraints and distributions of stimuli that could reduce the dimensionality of observed dynamics. We now emphasize this in the results section 'Global dynamical properties of the putative effectome':

We note the dimensionality of neural dynamics depends on the input distribution. This analysis implicitly assumes private white noise inputs to each neuron where all eigenvectors are equally likely to be driven---intuitively, this may correspond to a resting state with each neuron stochastically firing at a similar rate.

Referee #3 (Remarks on code availability):

Unfortunately I haven't had the time to review the code, I did check out the repo and the output figures though.